# The individual determinants of morning dream recall
Valentina Elce [1], Damiana Bergamo [1,2], Giorgia Bontempi [1], Bianca Pedreschi [1], Michele Bellesi [3], Giacomo Handjaras [1] & Giulio Bernardi [1] ✉

Evidence suggests that (almost) everyone dreams during their sleep and may actually do so for a large part of the night. Yet, dream recall shows large interindividual variability. Understanding the factors that influence dream recall is crucial for advancing our knowledge regarding dreams' origin, significance, and functions. Here, we tackled this issue by prospectively collecting dream reports along with demographic information and psychometric, cognitive, actigraphic, and electroencephalographic measures in 217 healthy adults (18–70 y, 116 female participants, 101 male participants). We found that attitude towards dreaming, proneness to mind wandering, and sleep patterns are associated with the probability of reporting a dream upon morning awakening. The likelihood of recalling dream content was predicted by age and vulnerability to interference. Moreover, dream recall appeared to be influenced by night-by-night changes in sleep patterns and showed seasonal fluctuations. Our results provide an account for previous observations regarding inter- and intra-individual variability in morning dream recall.

Dreams are subjective conscious experiences generated by the brain during sleep—when individuals are largely (though not completely[1]) disconnected from the external environment on the sensory (input) and motor (output) sides and are typically unable to exert volition and self-reflection[2]. Dream experiences draw on previously acquired memories and beliefs and, thus, present relevant aspects of continuity with thoughts, concerns, and salient experiences of our waking self[3,4]. In light of this, they are believed to represent an important window on—and to potentially have a direct role in—sleep-dependent processes involving learning and memory consolidation[5,6]. Moreover, dreams have a tight relationship with psycho-physical health[7]. In fact, alterations in the frequency or content of oneiric experiences may accompany, or even precede, the waking manifestation of clinical symptoms related to psychiatric and neurological disorders[7–9]. Finally, the study of dreaming and dreamless sleep is regarded as a fundamental experimental model in the search for the functional bases of human consciousness[10,11]. Indeed, as compared to task-based protocols exploring wakefulness-conscious experiences, the study of dreams is naturally less influenced by confounding effects such as changes in attention, stimulus and task processing, task performance, and response preparation[10,11].

In the 1950s, with the discovery of rapid eye movement (REM) sleep, researchers initially thought to have identified the neural correlates of dreaming[12,13], as dream experiences appeared to be far more common in this stage than in non-REM (NREM) sleep. This idea fits well with the fast, low-amplitude electroencephalographic (EEG) activity similar to wakefulness that characterizes REM sleep, as opposed to the slow, high-amplitude activity of NREM sleep. However, later studies partially amended this view. Indeed, serial-awakening laboratory investigations determined that contentful dreams are reported on average following ~85% of the awakenings from REM sleep and ~45% of the awakenings from NREM sleep (e.g., ref. 14).

While the sleep stage preceding the awakening is considered a key determinant for whether or not a dream will be reported, evidence indicates that dream recall probability fluctuates greatly both within and across individuals[15]. Such a variability attracted public and scientific attention during the recent pandemic, when an abrupt surge in morning dream recall was reported worldwide[16]. Yet, our current understanding of the factors influencing dream generation and recall is scarce. For instance, while several studies found female sex[17], younger age[18], a positive attitude towards dreaming, frequent daydreaming, and fantasy proneness, as consistently associated with a higher dream recall frequency[19–21], other investigations produced partially inconsistent or contradictory results (e.g., ref. 22). Results concerning the possible involvement of other personality or cognitive factors, such as visual and verbal memory, produced even more inconsistent results with some studies indicating a positive association[23,24] and others observing no significant predictive power[25,26]. These inconsistencies could be explained by differences in employed definitions and applied experimental approaches across studies (for detailed reviews, see refs. 15,27). Indeed, the available evidence is mostly based on retrospective measures potentially

[1]MoMiLab Research Unit, IMT School for Advanced Studies Lucca, Lucca, Italy. [2]Department of General Psychology, University of Padova, Padova, Italy. [3]School of Biosciences and Veterinary Medicine, University of Camerino, Camerino, Italy. ✉e-mail: giulio.bernardi@imtlucca.it

affected by biases such as memory- and personality-related distortions. Prospective studies conducted so far are sparse and hampered by significant limitations as, due to their higher costs, these investigations were typically performed on relatively small samples and explored only one or few variables potentially affecting dream recall (see refs. 28–30 for a discussion about pros and cons of prospective and retrospective approaches).

This picture is further complicated by the inherent foundation of dream studies, to some degree, on the assumption that reports provided by individuals upon awakening are a reliable reflection of dream occurrence and content[31]. However, any generated dream must be encoded in memory, and such a memory has to be later retrieved during wakefulness in order for a dream experience to be successfully recalled[15]. This issue is of particular importance given that memory processes appear to be altered during sleep and the subsequent period of *sleep inertia*. Indeed, individuals often wake up with the distinct feeling of having been dreaming but are unable to recall any detail of their experience. In some cases, the memory of the dream may be present at the moment of awakening but is rapidly lost if the experience is not immediately reported. These so-called "*white dreams*" have been interpreted as reflecting a failure of memory encoding or retrieval[32,33]. Yet, previous investigations provided little or no support for a direct relationship between memory skills and dream content recall[23–26].

Here we set out to investigate the intra- and inter-individual factors associated with morning dream recall in a large multimodal database collecting dream reports along with demographic information and psychometric, cognitive, actigraphic, and EEG measures. In this prospective, exploratory study, a cohort of healthy adults recorded a report of their last dream experience each morning upon spontaneous awakening at home for 15 days (Fig. 1a). Sleep–wake patterns were tracked through actigraphy. A subsample also wore a portable EEG device at night. Moreover, all participants were characterized across a wide range of cognitive and psychological dimensions.

## Methods

### Participants

The study was conducted on a sample of 217 healthy Italian native language speakers from 18 to 70 years old (116 female participants, 101 male participants). Of these, ten failed to comply with the experimental protocol, and three provided less than seven recordings (see below) leading to a final sample of 204 participants. Data collection was carried out between March 2020 and March 2024, covering a period of 4 years. Participants were recruited through word of mouth and the dissemination of virtual and paper flyers. Given the risks of recruiting a majority of volunteers with a specific interest in dreams, we mainly relied on word of mouth to reach diverse participants regardless of their interest towards dreaming. Only individuals with regular sleep/wake patterns, 6–8 h of sleep per night, and no diagnosis of sleep-related problems or of any other pathological condition that might have compromised their sleep were recruited in the study. Moreover, we excluded volunteers who were taking medications that could have affected sleep patterns at the time of the study and individuals who had a recent (last 6 months) history of alcohol and drug abuse. Finally, female volunteers who were pregnant, were planning a pregnancy, or were breastfeeding at the time of the study were also excluded.

Each study participant went through three phases (Fig. 1a): (i) a screening interview followed by the completion of a questionnaire battery, (ii) an experimental stage where sleep patterns and morning reports of subjective sleep-conscious experiences were collected for 15 days, and (iii) a final session with the administration of a battery of cognitive tests.

The study was conducted under a protocol approved by the Local Joint Ethical Committee for Research (#11/2020). All volunteers signed a written informed consent form before taking part in the study and retained the faculty to drop from the study at any time. Aims and analyses of the study were not preregistered.

**Fig. 1 | Description of the experimental protocol and collected data. a** Outline of the experimental paradigm. **b** Proportion of no dream experience (ND), white dream (WD), and contentful dream (CD) reports. For each report type, the corresponding raincloud plot (individual data points and probability distribution) and box plot are shown. On each box, the central mark indicates the median, and the bottom and top edges of the box indicate the 25th and 75th percentiles, respectively. The whiskers extend to the most extreme data points not considered outliers. **c** Correlation (Spearman's correlation coefficient) between demographic, psychological and cognitive variables derived from questionnaires and tests. Black dots indicate significant associations (q < 0.05, FDR correction; N = 204 participants). A moderate significant correlation was found between age and vulnerability to interference (r = −0.46). Relatively small but significant correlations also emerged between trait anxiety and subjective sleep quality (r = 0.28), proneness to mind wandering (r = 0.20), vividness of visual imagery (r = −0.21), chronotype (r = 0.20), and vulnerability to interference (r = 0.19), between attitude towards dreaming and education (r = −0.24), between chronotype and age (r = 0.26), between chronotype and vulnerability to interference (r = −0.21), as well as between verbal memory and sex (r = −0.19). Also see Supplementary Fig. 2.

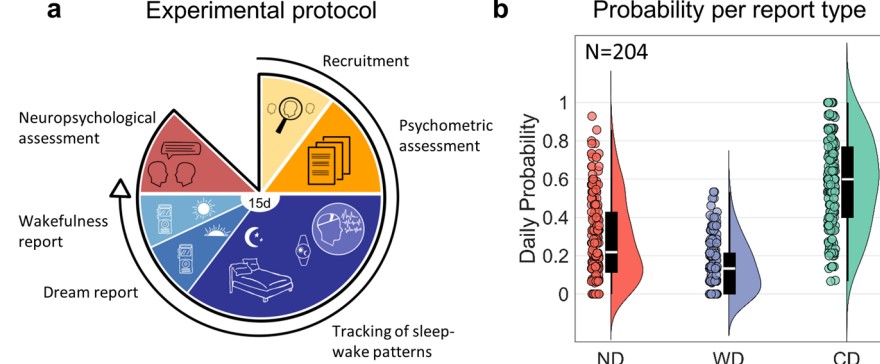

a Experimental protocol

b Probability per report type

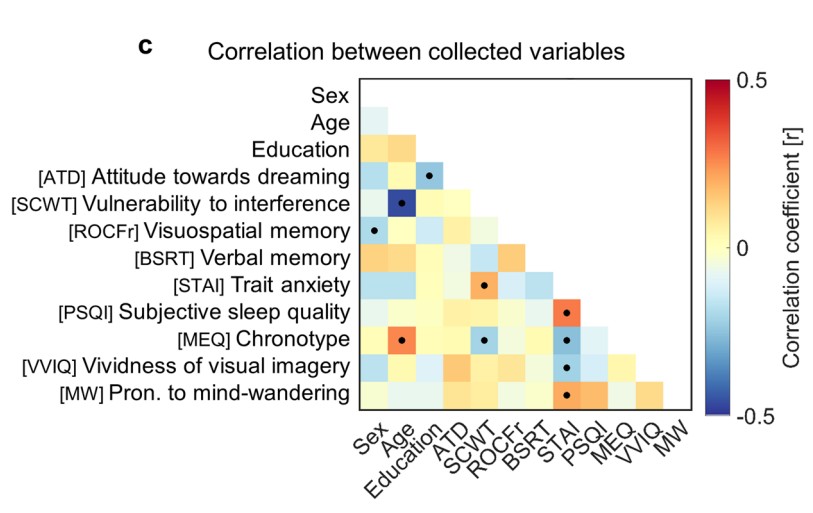

c Correlation between collected variables

### Screening interview and self-assessment questionnaires

All volunteers underwent an anamnestic interview aimed at assessing their general health and adherence to inclusion/exclusion criteria. Sex was determined by self-report. Recruited participants were then asked to fill out several questionnaires aimed at investigating their attitude towards dreaming[34], trait anxiety levels (*State-Trait Anxiety Inventory, STAI*[35]), vividness of visual imagery (*Vividness of Visual Imagery Questionnaire, VVIQ*[36]), proneness to mind wandering (*Mind Wandering—Spontaneous and Deliberate Scale, MW*[37]), subjective sleep quality (*Pittsburgh Sleep Quality Index, PSQI*[38]), subjective circadian preference (*Morningness-Eveningness Questionnaire*[39]). Attitude towards dreaming was assessed using a 6-item questionnaire where participants were asked to provide their degree of agreement with six statements regarding the general meaning and significance of dreams on a Likert Scale from 0 ("completely disagree") to 4 ("completely agree"). Three items were positive statements about dreams (e.g., "dreams are a good way of learning about my true feelings") and three were negative (e.g., "dreams are random nonsense from the brain"). A global score was computed by subtracting the sum of scores provided to the negative statements from the sum of scores associated with the positive statements. Finally, participants completed a questionnaire about their dream experiences in the previous three months, which included one item aimed at assessing the frequency of morning dream recall[40].

### Collection of morning dream reports and sleep patterns

Volunteers who met all the inclusion criteria were provided with an actigraph and a voice recorder and were asked to record each morning, upon awakening from sleep, everything that was going through their mind just before they woke up, everything they remembered, every experience or thought they had before awakening. It is important to note that this procedure differed in several respects from those commonly used in home-based dream experiments. First, all participants received clear, standardized instructions on what to consider as a dream. In particular, we adopted a broad definition, encompassing any subjective conscious experience occurring during sleep[41]. Second, while in some studies participants are asked to report all dreams that they have during a given sleep night, here we instructed them to only focus on the very last experience that they had before awakening. This choice was made in order to minimize the impact of confounding effects that may intervene between the dream experience and the report. Third, while most previous studies relied on written reporting, here we used voice recordings, as this approach renders the recall task simpler and less demanding for participants[42]. Furthermore, at pseudo-random times during the day, participants were also contacted and asked to record everything that was going through their minds up to 15 min before they started the recording. In particular, a simple phone text message containing the word "record" ("*registra*") was sent to the volunteers at pseudo-random times during the day (Fig. 1a). Wakefulness reports were not analyzed in this study.

Only participants who provided at least seven reports during the experimental period were included in our analyses. This selection criterion led to the exclusion of three participants. One participant who reported altered sleep–wake patterns (sleep restriction) in three nights voluntarily extended the experimental period to 20 days. We discarded the reports provided by this participant at the awakening from the altered nights and on the experimental days immediately following. Furthermore, we excluded delayed dream recall reports. Though volunteers were specifically required to only report the experiences they remembered right after the awakening, occasionally they retrieved and recorded the dream experience later during the day. Since these memories might be triggered and distorted by external stimuli and events experienced during wakefulness, we chose to exclude those data.

During the 15 days of the study, participants wore an actigraph to track sleep–wake patterns (MotionWatch-8, Camtech). A subgroup of 50 volunteers (27 female participants, 23 male participants; age 29.7 ± 5.2 years, range 22–44 years) also had their sleep-related brain activity recorded through a portable EEG system (*DREEM*) equipped with five EEG dry electrodes (seven derivations: Fp1-O1, Fp1-O2, Fp1-F7, F8-F7, F7-O1, F8-O2, Fp1-F8), a pulse sensor, and a 3D accelerometer. Eight participants interrupted EEG data collection due to discomfort while sleeping. Therefore, we were able to analyze data collected from 42 participants.

### Cognitive testing

At the end of the two-week period, all participants underwent a neuropsychological assessment aimed at evaluating different cognitive abilities. The neuropsychological tests comprised: *Stroop Color and Word Test*, for assessing participants' processing speed and vulnerability to cognitive interference (*SCWT*[43]); *Babcock Story Recall Test* (immediate recalling—delayed recalling), for evaluating participants' episodic and verbal memory (*BSRC*[44]); *Rey–Osterrieth complex figure* (immediate copy—delayed copy), for evaluating participants' visuo-constructional ability and visual memory (*ROCFr*[45]).

### Analysis of actigraphic data

Actigraphic recordings were evaluated by means of the *MotionWare* Software[46]. The following 22 measures were computed (as described within the software user guide): *actual sleep (or wake) time* (the total time spent in sleep/wake according to the epoch-by-epoch wake/sleep categorization); *actual sleep (or wake) percent* (actual sleep/wake time expressed as a percentage of the total elapsed time between "*fell asleep*" and "*wake up*" times); *sleep efficiency* (actual sleep time expressed as a percentage of time in bed); *sleep (or wake) bouts* (the number of contiguous sections categorized as sleep/wake in the epoch-by-epoch wake/sleep categorization); *mean sleep (or wake) bout* (the average length of each of the sleep/wake bouts); *immobile (or mobile) minutes* (the total time categorized as immobile/mobile in the epoch-by-epoch mobile/immobile categorization); *percentage of immobile (or mobile) time* (the immobile/mobile time expressed as a percentage of the assumed sleep time); *immobile bouts* (the number of contiguous sections categorized as immobile in the epoch-by-epoch mobile and immobile categorization); *mean immobile bout* (the average length of each of the immobile bouts); *immobile bouts <=1 min* (the number of immobile bouts which were less than or equal to 1 min in length); *percentage of immobile bouts <=1 min* (the number of immobile bouts less than or equal to 1 min expressed as a percentage of the total number of immobile bouts); *total activity score* (the total of all the activity counts during the assumed sleep period); *mean activity/epoch* (the total activity score divided by the number of epochs in the assumed sleep period); *mean nonzero activity per epoch* (the total activity score divided by the number of epochs with greater than zero activity in the assumed sleep period); *fragmentation index* (the sum of the "*percentage of mobile time*" and the "*percentage of immobile bouts <=1 min*"); *central phase measure* (the midpoint between the "*fell asleep*" and "*wake up*" times, expressed as the number of minutes past midnight). Moreover, we expressed the '*fell asleep*' and the "*wake up*" times as the number of minutes past midnight.

We discarded measures from single nights where the actigraph appeared to have been removed (e.g., cases where participants removed it and forgot to wear it again before sleep time). In particular, we discarded nights that met at least three of the following heuristic criteria: number of "*immobile bouts <=1 min*" ≤ 2, number of "*sleep bouts*" ≤ 4, "*fragmentation index*" ≤ 1.5%, "*actual sleep percent*" ≥ 96%. Overall, we excluded actigraphic recordings collected in 39 nights across 28 participants due to missing or unreliable data. Moreover, actigraphic data (all nights) was lost in four participants due to technical issues (4 female participants, age 22–30 years).

Given that the actigraphic variables were highly correlated with each other, we applied dimensionality reduction through principal component analysis (PCA; N = 2845 nights). We retained PCs that explained at least 10% of the total variance. The PCA was applied by combining all actigraphic data from different nights across the entire sample of participants, effectively mixing intra- and inter-individual variability. In order to rule out potential biases in the generation of the PCA loadings, we repeated the analysis using Multiple Factor Analysis (MFA[47]). MFA is a generalization of PCA designed for data organized into multiple blocks, such as different sets of variables collected from the same observations or, as in our case, the same variables measured across different observations[48]. We found that the two techniques

produced highly similar loadings across the first four PCs, with average correlations across nights of 0.995 ± 0.004 for PC1, 0.976 ± 0.018 for PC2, 0.986 ± 0.010 for PC3, and 0.924 ± 0.028 for PC4 (Supplementary Fig. 1). Overall, this evidence demonstrates that intra-individual variability did not introduce any bias in our estimation of PC loadings and scores.

### Analysis of portable EEG system data

Data collected using the *DREEM* device were analyzed using the associated automated sleep scoring software[49]. Of note, one experimenter manually inspected the sleep scoring output and eliminated nights containing obvious issues related to possible device removal or malfunction. Moreover, we excluded nights for which less than 5 h of sleep were recorded, and nights in which more than 25% of all the epochs were marked as unscorable. The obtained hypnograms were used to compute the percentages of wakefulness, N1, N2, N3, and REM sleep for each night ($N = 480$). Then, sleep structure measures were used to facilitate the interpretation of actigraphy-related PCs. Specifically, we employed mixed-effect models to explore the association between sleep structure measures and each of the four PCs. Four identical, independent models were used. An FDR correction[50] for multiple comparisons was applied to adjust $p$ values assigned to each of the tested predictors.

Below we reported the adopted model using Wilkinson's notation, where Y represents the predicted variable (i.e., each PC), Subj is the participant identification number and Night is the experimental night counted from the beginning of the experiment:

$$Y \sim Sex + Age + \%W + \%N1 + \%N2 + \%N3 + \%REM + (1|Subj) + (-1 + Night|Subj)$$

### Predictors of dream recall and dream content memory

All morning verbal reports were evaluated and classified as either contentful dream experience (CD) if the verbal description included at least one reference to any kind of semantic content, dream experience without recall of content ("*white dream*," WD) if the participant reported the perception of having dreamt but could not recall any feature of the experience, and no dream experience (ND) if the participant woke up with the feeling of not having dreamt. Moreover, the sum of dream experiences with and without recall of content (CD + WD) was computed to obtain an estimate of all cases where participants reported having dreamt. We assumed this metric to represent the best measurable estimate of generated dreams.

Two separate mixed-effect logistic models were used to investigate the inter-individual predictors of dream recall (CD + WD vs. NE) and dream content memory (CD vs. WD). The same predictors were included in the two models: age, sex, education, attitude towards dreaming (ATD), trait anxiety (STAI), vulnerability to interference (SCWT), vividness of visual imagery (VVIQ), proneness to mind wandering (MW), verbal memory (BSRT), visual memory (ROCFr), subjective sleep quality (PSQI), subjective circadian preference (MEQ), and four actigraphy-derived PCs (see "Results"). The models also accounted for possible effects of the experimental days when the reports were collected.

The first model, aimed at predicting morning dream recall (CD + WD) as compared to ND reports, included a total of 2900 reports across 204 participants. The second model, aimed at predicting contentful dream recall (CD) as compared to WD reports, included a total of 2077 reports across 204 participants. An FDR correction[50] for multiple comparisons was applied to adjust $P$ values assigned to each of the tested predictors.

Similarly to the analysis of EEG data, Subj and Night variables were included as random-effect terms. Below is the model used:

$$Y \sim Sex + Age + Education + ATD + MW + ROCFr + BSRT + VVIQ + SCWT + PSQI + MEQ + STAI + PC1 + PC2 + PC3 + PC4 + (1|Subj) + (-1 + Night|Subj)$$

### Night-by-night variations in dream recall

We investigated how sleep patterns affect night-by-night variations in morning dream recall. Similar analyses were performed using actigraphy-derived PC scores and sleep structure measures obtained from EEG data. Specifically, we first averaged values for nights associated with a dream report (CD + WD) and nights that were not followed by a dream report (ND). Then mean values were compared across report types using non-parametric sign-rank tests for paired samples. In case a significant effect was found for a specific PC or sleep structure parameter, additional comparisons were carried out across CD, WD, and NE report types. FDR corrections were applied to account for multiple comparisons.

### Seasonal variations in dream recall

To investigate the impact of seasonal variations on morning dream recall, we estimated morning dream recall probability per participant and adjusted the obtained values for age, sex, attitude towards dreaming, proneness to mind wandering, and long light sleep (PC2) scores (see "Results"). Then, we grouped volunteers according to the season (Winter, Spring, Summer, Autumn) in which they carried out the study, using as a reference the central day of their experimental period. Rank-sum tests were performed to compare adjacent seasons, and an FDR correction was applied to account for multiple comparisons.

To determine whether the seasonal changes in dream recall could be explained by changes in sleep patterns, we analyzed potential seasonal variations in actigraphy-derived PC scores. For these analyses, mean PC scores were independently adjusted for age and sex. Possible effects were assessed as described above.

### Reporting summary

Further information on research design is available in the Nature Portfolio Reporting Summary linked to this article.

## Results

A total of 204 participants were sampled from the healthy Italian adult population (mean age 35.1 ± 12.5 years; 113 female participants, 91 male participants), and 2900 morning reports were included in the analyses (14.22 ± 1.44 reports per participant). Forty-two participants wore a portable EEG device during their sleep (24 female participants, 18 male participants; age 30.0 ± 5.2 years, range 22–44 years).

### Frequency of morning dream reports

Figure 1b shows the proportion of morning reports corresponding to CD (mean ± standard deviation = 0.58 ± 0.24), WD (0.14 ± 0.13), and ND (0.28 ± 0.22). On average, CD + WD probability was 0.72 ± 0.22, corresponding to 5.04 ± 1.54 dreams per week. Morning dream recall frequency computed from the verbal diary was significantly higher than self-reported dream recall frequency (2.66 ± 2.29; signed-rank test, $P < 0.0001$; $|g| = 1.19$, CI = [1.05, 1.36]). The two indices of dream recall showed a moderate positive correlation (Spearman's correlation, $P < 0.0001$; $r = 0.46$, CI = [0.35, 0.56]). These findings are consistent with previous studies indicating an incomplete correspondence between retrospective and prospective measures of dream recall[15].

### Inter-individual predictors

Next, we performed mixed-effect logistic regression analyses to identify potential predictors of dream generation (CD + WD vs. ND) and dream content memory (CD vs. WD). The following predictors were selected based on previous literature: age, sex, years of education, attitude towards dreaming, vulnerability to cognitive interference, verbal memory, visuospatial memory, trait anxiety, vividness of visual imagery, proneness to mind wandering, self-reported sleep quality, and self-reported chronotype (relative correlations among these predictors are shown in Fig. 1c). Moreover, objective sleep measures were derived from actigraphic data. In particular, a

principal component analysis (PCA) was performed across 24 distinct actigraphic indices (see "Methods"). We obtained four PCs together explaining 87.74% of the total variance (PC1: 50.8%; PC2: 15.4%; PC3: 11.2%; PC4: 10.4%; Fig. 2). To facilitate the interpretation of the PCs, we employed mixed-effect models including sleep structure measures obtained in the subsample of participants who wore the portable EEG system during the experimental nights (Tables 1–4).

We found that PC1 was positively associated with the proportion of wakefulness ($q = 0.001$, False Discovery Rate—FDR- correction) and N1 ($q = 0.001$; model adjusted $R^2 = 0.47$). Moreover, PC2 was negatively associated with the proportion of N3 sleep ($q < 0.001$; model adjusted $R^2 = 0.41$), whereas PC4 was negatively associated with the proportions of N2, N3, and REM sleep ($q < 0.005$; model adjusted $R^2 = 0.38$). No significant predictors were identified for PC3. Based on these observations (Fig. 2b) and the distribution of PC loadings (Fig. 2a), the four PCs could be assumed to

mainly reflect, respectively, *sleep fragmentation* (PC1), *long, non-N3 sleep* (PC2; hereinafter indicated as "long light sleep"), *stable sleep with advanced phase* (PC3; "stable advanced sleep"), and *unstable sleep with advanced phase* (PC4; "unstable advanced sleep").

We found that morning dream reports were predicted by attitude towards dreaming, proneness to mind wandering, and long light sleep ($q < 0.05$, FDR correction; adjusted $R^2 = 0.17$; Fig. 3, also see Table 5).

Contrary to previous research[17,51], we did not find significant effects of age and sex on dream recall (i.e., higher recall in younger individuals and female individuals). However, we noted a significant relationship between sex and attitude towards dreaming, with the latter being higher in female volunteers ($N = 113$, $36.6 \pm 13.4$ years) compared to male volunteers ($N = 91$, $33.3 \pm 11.2$ years; rank-sum test, $P = 0.014$; |g| = 0.34, CI = [0.07, 0.63]). Moreover, we found a significant negative correlation between light sleep (PC2) and age (Spearman's correlation, $P = 0.008$; $r = -0.19$, CI =

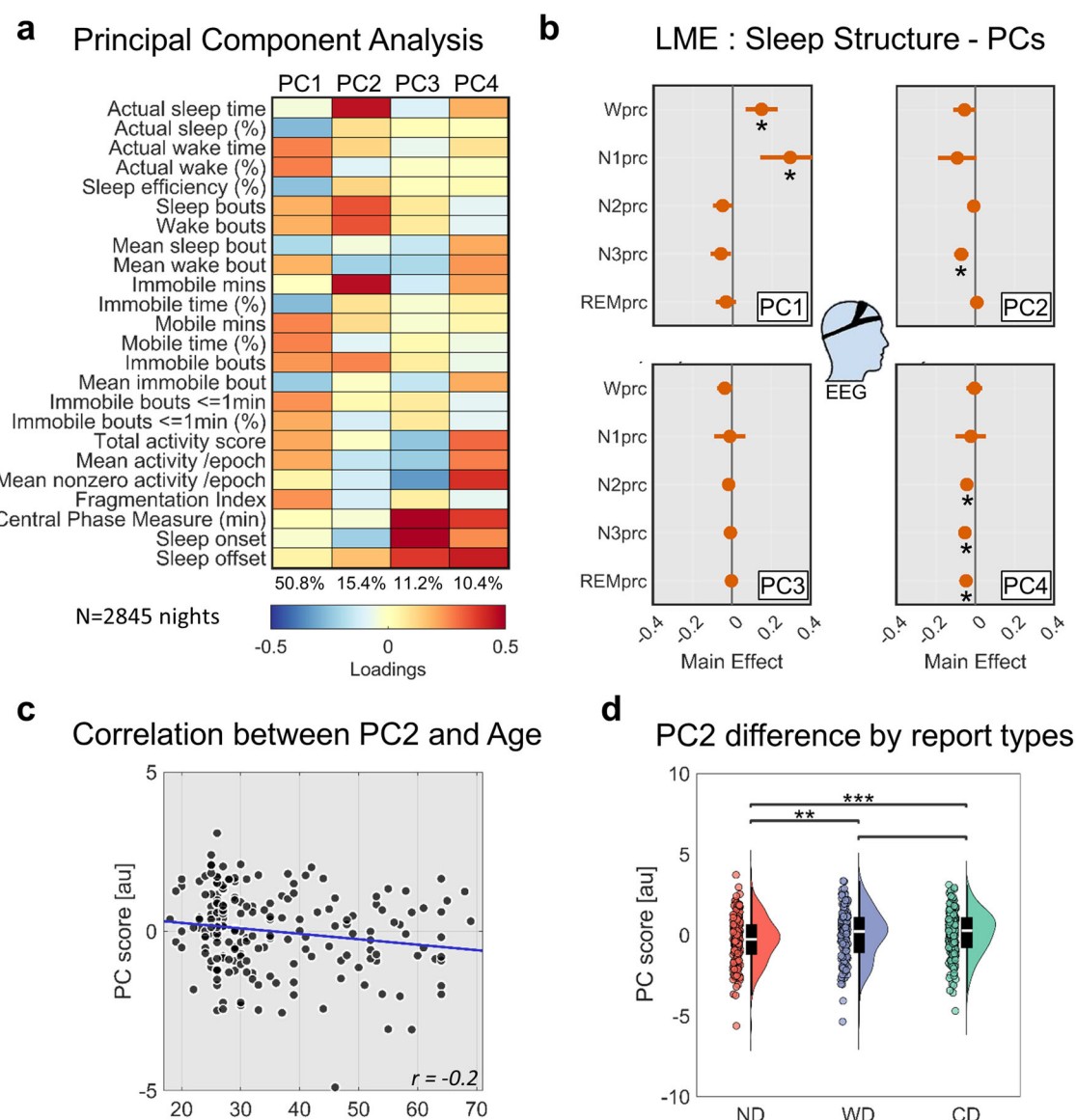

**Fig. 2 | Analysis of the sleep patterns. a** Principal component (PC) analysis of actigraphic data ($N = 2845$ nights, 200 participants). We identified four PCs explaining at least 10% of the variance. For each PC, the plot shows the loadings on each actigraphic metric and the explained variance (bottom). **b** Linear mixed-effect (LME) models were applied to investigate the EEG-derived sleep structure parameters associated with each PC (* marks significant effect at $q < 0.05$, FDR

correction; $N = 480$ nights, 42 participants). Age and sex were accounted for in the models, but their effects are not shown. **c** Correlation between PC2 loadings and age ($N = 200$ participants). A significant negative correlation was observed.
**d** Comparison of mean PC2 loadings associated with nights followed by no dream experience (ND), white dream (WD), and contentful dream (CD) reports (*$P < 0.05$; **$P < 0.01$; ***$P < 0.001$; $N = 200$ participants).

## Table 1 | LME results for PC1 (df = 448)

| Name | Estimate | SE | tStat | *P* value | Lower CI | Upper CI |
|---|---|---|---|---|---|---|
| Intercept | 2.7179 | 2.7305 | 0.99536 | 0.3201 | −2.6483 | 8.0841 |
| Sex | 0.96943 | 0.52753 | 1.8377 | 0.066773 | −0.067315 | 2.0062 |
| Age | −0.041004 | 0.051447 | −0.797 | 0.42587 | −0.14211 | 0.060104 |
| Wprc | 0.14616 | 0.041121 | 3.5544 | 0.00041912 | 0.065346 | 0.22698 |
| N1prc | 0.29076 | 0.077187 | 3.767 | 0.00018729 | 0.13907 | 0.44245 |
| N2prc | −0.050788 | 0.024653 | −2.0601 | 0.039964 | −0.099238 | −0.0023383 |
| N3prc | −0.05986 | 0.02611 | −2.2926 | 0.022335 | −0.11117 | −0.0085458 |
| REMprc | −0.034066 | 0.025822 | −1.3193 | 0.18776 | −0.084814 | 0.016682 |

For each predictor, we reported the estimate, standard error (SE), t statistics, *P* value, and lower and upper bounds of the effect 95% confidence interval.

## Table 2 | LME results for PC2 (df = 448)

| Name | Estimate | SE | tStat | *P* value | Lower CI | Upper CI |
|---|---|---|---|---|---|---|
| Intercept | 4.4653 | 1.8226 | 2.45 | 0.014666 | 0.8835 | 8.0472 |
| Sex | −0.002238 | 0.31624 | −0.0070768 | 0.99436 | −0.62373 | 0.61925 |
| Age | −0.065779 | 0.030879 | −2.1302 | 0.0337 | −0.12646 | −0.0050924 |
| Wprc | −0.054373 | 0.028667 | −1.8967 | 0.058505 | −0.11071 | 0.0019646 |
| N1prc | −0.091105 | 0.050223 | −1.814 | 0.070343 | −0.18981 | 0.0075962 |
| N2prc | −0.0079423 | 0.017263 | −0.46007 | 0.64569 | −0.041869 | 0.025984 |
| N3prc | −0.071085 | 0.018227 | −3.8999 | 0.00011097 | −0.10691 | −0.035264 |
| REMprc | 0.0079423 | 0.018042 | 0.44022 | 0.65999 | −0.027515 | 0.043399 |

For each predictor, we reported the estimate, standard error (SE), t statistics, *P* value, and lower and upper bounds of the effect 95% confidence interval.

## Table 3 | LME results for PC3 (df = 448)

| Name | Estimate | SE | tStat | *P* value | Lower CI | Upper CI |
|---|---|---|---|---|---|---|
| Intercept | 3.3489 | 1.4277 | 2.3457 | 0.019427 | 0.54313 | 6.1547 |
| Sex | 0.2693 | 0.3165 | 0.85088 | 0.39529 | −0.3527 | 0.8913 |
| Age | −0.075295 | 0.03081 | −2.4438 | 0.014917 | −0.13585 | −0.014745 |
| Wprc | −0.036081 | 0.019798 | −1.8224 | 0.069055 | −0.074989 | 0.0028279 |
| N1prc | −0.0095915 | 0.040251 | −0.23829 | 0.81176 | −0.088696 | 0.069513 |
| N2prc | −0.016468 | 0.011812 | −1.3942 | 0.16394 | −0.039681 | 0.0067449 |
| N3prc | −0.0072681 | 0.012547 | −0.57927 | 0.5627 | −0.031926 | 0.01739 |
| REMprc | −0.0021302 | 0.012398 | −0.17182 | 0.86366 | −0.026495 | 0.022235 |

For each predictor, we reported the estimate, standard error (SE), t statistics, *P* value, and lower and upper bounds of the effect 95% confidence interval.

## Table 4 | LME results for PC4 (df = 448)

| Name | Estimate | SE | tStat | *P* value | Lower CI | Upper CI |
|---|---|---|---|---|---|---|
| Intercept | 5.5712 | 1.3946 | 3.9949 | 7.56E-05 | 2.8305 | 8.312 |
| Sex | −0.4531 | 0.26137 | −1.7336 | 0.083678 | −0.96676 | 0.060552 |
| Age | −0.042873 | 0.025499 | −1.6814 | 0.093388 | −0.092985 | 0.0072392 |
| Wprc | −0.0047039 | 0.021293 | −0.22091 | 0.82526 | −0.046551 | 0.037143 |
| N1prc | −0.022479 | 0.039219 | −0.57316 | 0.56682 | −0.099554 | 0.054597 |
| N2prc | −0.042706 | 0.012781 | −3.3414 | 0.00090348 | −0.067824 | −0.017588 |
| N3prc | −0.052609 | 0.013526 | −3.8895 | 0.00011568 | −0.079191 | −0.026027 |
| REMprc | −0.046647 | 0.01338 | −3.4864 | 0.00053782 | −0.072942 | −0.020352 |

For each predictor, we reported the estimate, standard error (SE), t statistics, *P* value, and lower and upper bounds of the effect 95% confidence interval.

**Fig. 3 | Predictors of dream recall and memory.** Linear mixed-effect models exploring the inter-individual predictors of morning dream recall (**a**; N = 2900) and dream content memory (**b**; N = 2077). The overall effects are shown for each variable included in the models. *Mark significant effects at q < 0.05, FDR correction.

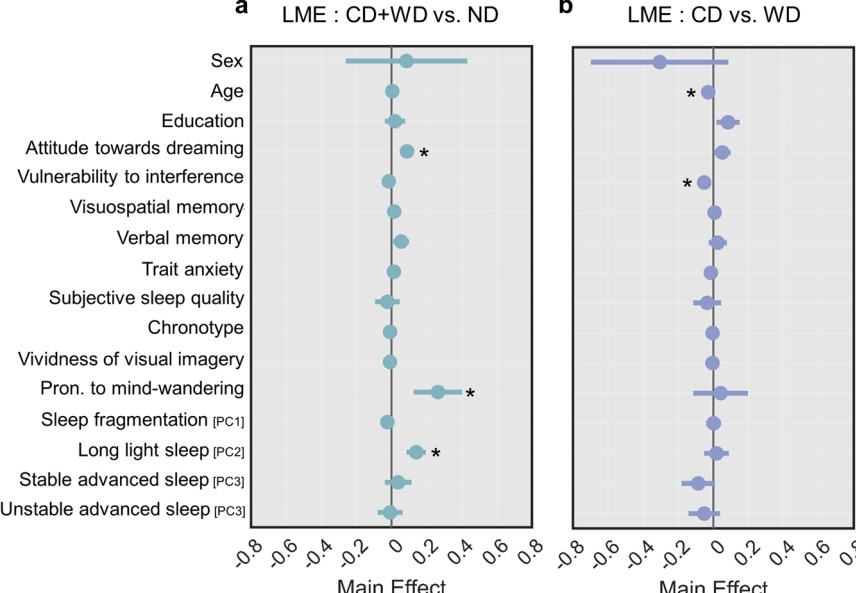

Linear Mixed-Effect Models – Predictors of dream recall and dream memory

**a** LME : CD+WD vs. ND

**b** LME : CD vs. WD

[−0.29, −0.11]). These results suggest that previously described effects of age and sex could have been mediated by other factors. The distinction between CD and WD was instead predicted by age and vulnerability to interference (q < 0.05; adjusted $R^2$ = 0.11; Fig. 3b, also see Table 6).

### Night-by-night variations

Next, we investigated how sleep patterns affect night-by-night variations in morning dream recall. We thus compared mean actigraphic PCs for nights followed (CD + WD) or not (ND) by a morning dream report. No significant differences were found for sleep fragmentation (PC1; P = 0.101, uncorrected; N = 182), stable advanced sleep (PC3; P = 0.932), and unstable advanced sleep (PC4; P = 0.918). Instead, long light sleep (PC2) scores were significantly higher for CD + WD relative to ND (N = 182; corrected q < 0.001; |g| = 0.28 − [0.17, 0.41]). Therefore, we further investigated for this PC potential differences among CD, WD, and ND. We found that both WD (N = 129; q = 0.005; |g| = 0.22, CI = [0.08, 0.38]) and CD (N = 182; q < 0.001; |g| = 0.26, CI = [0.14, 0.38]) had significantly higher scores relative to ND, whereas no differences were found between WD and CD (N = 139; P = 0.681, uncorrected). Overall, these results indicate that CD and WD depend on similar sleep patterns and suggest that long light sleep may be associated with dreaming per se, rather than dream content memory.

In addition, we investigated the impact of sleep structure measures on dream recall in the sample of participants who wore the portable EEG system. We found that morning dream recall tended to be associated with the proportion of overnight REM sleep (N = 37; uncorrected P = 0.025), but this effect did not survive correction for multiple comparisons (q = 0.125; |g| = 0.37, CI = [0.04, 0.74]). A follow-up contrast among CD, WD, and ND reports showed a significant difference in the amount of REM sleep between CD and ND (N = 37; uncorrected P = 0.023; corrected q = 0.070; |g| = 0.39 − [0.04, 0.74]) but not between WD and ND (N = 25; P = 0.313) or between CD and WD (N = 28; P = 0.466). It is important to note that this analysis concerned the sleep macrostructure of the entire night and that a longer REM duration does not necessarily imply that participants woke up from this stage. However, similar analyses performed using the last 2 h of sleep yielded similar results (CD + WD vs. ND, P = 0.018; CD vs. ND, P = 0.007; CD vs. WD, P = 0.029; WD vs. ND, P = 0.696).

Overall, the obtained results suggest that individuals may be more likely to recall dreams when they wake up from long sleep nights with a small proportion of deep, N3 sleep and higher REM content. This finding is

consistent with previous observations indicating a negative correlation between sleep stages with a high slow wave activity (N3) and dreaming[25,52].

### Seasonal variations

Owing to the fact that data collection took place over a period of 4 years (from 2020 to 2024), we investigated potential fluctuations in morning dream recall across seasonal cycles (Fig. 4). For this analysis, morning dream report (CD + WD) rates were computed for each participant and adjusted for age, attitude towards dreaming, proneness to mind wandering and mean light sleep (PC2) scores. We found that morning dream report probability was significantly lower in Winter relative to Spring (uncorrected P = 0.005, corrected q = 0.019; |g| = 0.59, CI = [0.19, 1.05]). A trend towards a lower morning dream recall in Winter relative to Autumn was also observed (uncorrected P = 0.039, corrected q = 0.077; |g| = 0.44, CI = [0.06, 0.80]).

To determine whether the observed seasonal changes could mirror changes in sleep patterns as assessed using actigraphic indices, we further analyzed potential seasonal variations in PC scores. For these analyses, mean PC scores were adjusted for age and sex. We found no significant seasonal changes for sleep fragmentation (PC1; uncorrected P > 0.074), stable advanced sleep (PC3; P > 0.184), and unstable advanced sleep (PC4; P > 0.184). A significant impact of the season was instead found for long light sleep (PC2). In particular, long light sleep scores were lower in Summer relative to both Spring (uncorrected P = 0.005, corrected q = 0.010; |g| = 0.60, CI = [0.21, 1.09]) and Autumn (uncorrected P < 0.001, corrected q < 0.001; |g| = 0.81, CI = [0.38, 1.32]). Overall, seasonal sleep changes did not appear to mirror relative changes in dream recall across seasons.

### Discussion

Here, we show that the likelihood of waking up in the morning from a subjective dream experience is predicted by three main factors, that are attitude towards dreaming, proneness to mind wandering, and trait differences in overnight sleep patterns. Moreover, individual differences in the tendency to recall the content of dream experiences as opposed to the mere awareness of having dreamt are predicted by vulnerability to interference and age.

A positive association between attitude towards dreaming and dream recall has been consistently described both by studies employing retrospective measures, such as questionnaires, and by prospective investigations performed, among other approaches, by means of dream diaries[27]. Yet, the

**Table 5 | LME results for CD + WD vs. ND (df = 2779)**

| Name | Estimate | SE | tStat | P value | Lower CI | Upper CI |
|---|---|---|---|---|---|---|
| Intercept | −1.0361 | 1.2414 | −0.83464 | 0.40399 | −3.4703 | 1.398 |
| Sex | 0.085477 | 0.17631 | 0.4848 | 0.62786 | −0.26024 | 0.4312 |
| Age | 0.0045574 | 0.0077018 | 0.59173 | 0.55408 | −0.010544 | 0.019659 |
| Education | 0.019654 | 0.029656 | 0.66275 | 0.50755 | −0.038495 | 0.077804 |
| ATD | 0.088294 | 0.021398 | 4.1262 | 3.80E-05 | 0.046336 | 0.13025 |
| SCWT | −0.015632 | 0.013154 | −1.1884 | 0.23479 | −0.041425 | 0.010161 |
| ROCFr | 0.015379 | 0.015884 | 0.96818 | 0.33304 | −0.015767 | 0.046525 |
| BSRT | 0.055183 | 0.023214 | 2.3771 | 0.017517 | 0.0096634 | 0.1007 |
| STAI | 0.013322 | 0.010258 | 1.2987 | 0.19417 | −0.0067928 | 0.033437 |
| PSQI | −0.023547 | 0.036135 | −0.65164 | 0.51469 | −0.094401 | 0.047307 |
| MEQ | −0.0084313 | 0.0086472 | −0.97503 | 0.32963 | −0.025387 | 0.0085243 |
| VVIQ | −0.0089557 | 0.0073665 | −1.2157 | 0.22419 | −0.0234 | 0.0054886 |
| MW | 0.26467 | 0.070481 | 3.7551 | 0.0001768 | 0.12647 | 0.40287 |
| PC1 | −0.023383 | 0.01649 | −1.418 | 0.1563 | −0.055717 | 0.0089507 |
| PC2 | 0.1408 | 0.027931 | 5.0409 | 4.93E-07 | 0.086031 | 0.19557 |
| PC3 | 0.037913 | 0.038789 | 0.97742 | 0.32845 | −0.038146 | 0.11397 |
| PC4 | −0.0078007 | 0.036396 | −0.21433 | 0.8303 | −0.079166 | 0.063565 |

For each predictor, we reported the estimate, standard error (SE), t statistics, P value, and lower and upper bounds of the effect 95% confidence interval.

**Table 6 | LME results for CD vs. WD (df = 1983)**

| Name | Estimate | SE | tStat | P value | Lower CI | Upper CI |
|---|---|---|---|---|---|---|
| Intercept | 2.8894 | 1.3872 | 2.0829 | 0.037392 | 0.16881 | 5.6099 |
| Sex | −0.30565 | 0.19979 | −1.5298 | 0.12622 | −0.69747 | 0.086176 |
| Age | −0.029499 | 0.0083988 | −3.5123 | 0.00045411 | −0.045971 | −0.013028 |
| Education | 0.083547 | 0.033256 | 2.5122 | 0.012076 | 0.018326 | 0.14877 |
| ATD | 0.050675 | 0.024354 | 2.0807 | 0.037586 | 0.0029122 | 0.098438 |
| SCWT | −0.052596 | 0.015131 | −3.476 | 0.00051982 | −0.08227 | −0.022922 |
| ROCFr | 0.0067291 | 0.017447 | 0.38569 | 0.69977 | −0.027487 | 0.040945 |
| BSRT | 0.024485 | 0.026375 | 0.92834 | 0.35334 | −0.02724 | 0.07621 |
| STAI | −0.014504 | 0.011792 | −1.23 | 0.21886 | −0.037631 | 0.0086225 |
| PSQI | −0.035278 | 0.04032 | −0.87495 | 0.38171 | −0.11435 | 0.043796 |
| MEQ | −0.0045697 | 0.010165 | −0.44955 | 0.65309 | −0.024505 | 0.015366 |
| VVIQ | −0.005 | 0.0081891 | −0.61056 | 0.54156 | −0.02106 | 0.01106 |
| MW | 0.040873 | 0.079046 | 0.51708 | 0.60516 | −0.11415 | 0.19589 |
| PC1 | 0.001104 | 0.021331 | 0.051757 | 0.95873 | −0.040729 | 0.042937 |
| PC2 | 0.017804 | 0.035843 | 0.49672 | 0.61944 | −0.05249 | 0.088097 |
| PC3 | −0.086444 | 0.04868 | −1.7758 | 0.075927 | −0.18191 | 0.0090257 |
| PC4 | −0.052475 | 0.045772 | −1.1464 | 0.25175 | −0.14224 | 0.037291 |

For each predictor, we reported the estimate, standard error (SE), t statistics, P value, and lower and upper bounds of the effect 95% confidence interval.

causal relationship still represents an open question. Indeed, it has been suggested that a pre-existing interest in dreams may drive a person to apply strategies aimed at increasing successful dream retrieval (e.g., keeping a dream diary). On the other hand, individuals who often remember their dreams may develop an interest in their possible meaning or significance. Notably, our findings indicate that while attitude towards dreams influences the likelihood of reporting the experience of a dream, it does not significantly impact the probability of recalling dream content. This observation lends indirect support to the notion that the association between attitude towards dreaming and dream recall may be driven by factors beyond mere memory processes.

The tendency towards mind wandering emerges in our study as another robust positive predictor of dream recall. A relevant perspective posits that dreaming and mind wandering, or daydreaming, may exist along a continuum, relying on similar brain functional mechanisms and structures[53–56]. Recent research has highlighted the involvement of over-lapping neural networks, particularly the default mode network (DMN), in both mind wandering and dreaming[54,57]. The DMN, including brain regions such as the medial prefrontal cortex and posterior cingulate cortex, is known to be active during periods of internally focused cognition and self-referential thought[58]. Given its role in introspective mental processes, the DMN has been implicated in promoting mind wandering during

**Fig. 4 | Seasonal changes in dream recall and sleep.** Changes in morning dream recall (**a**) and sleep patterns reflected by the actigraphy-PC2 (**b**) across seasons. *$q < 0.05$, **$q < 0.01$, ***$q < 0.001$ (FDR correction). Dream recall probability values were adjusted for age, sex, vulnerability to interference, attitude towards dreaming, and PC2. PC scores were adjusted for participants' age and sex. The analysis included 48 participants in Winter, 50 in Spring, 42 in Summer, and 60 in Autumn.

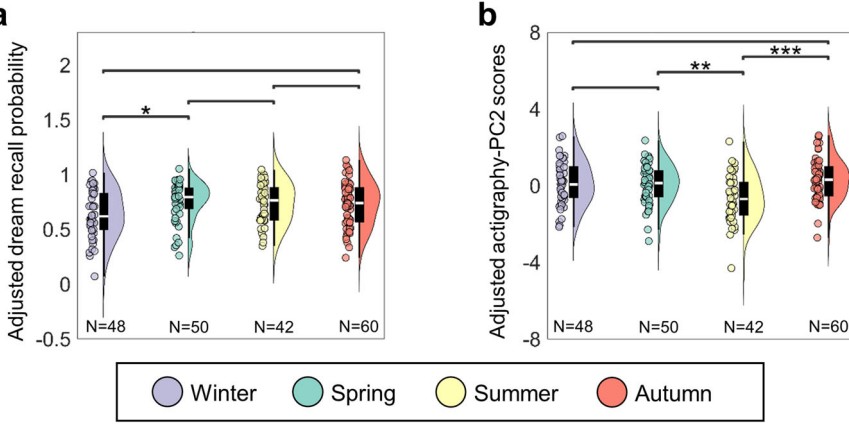

wakefulness[59,60]. Consequently, the association between mind wandering and dream recall observed in our findings may indicate a heightened propensity to spontaneously generate dream-like experiences, irrespective of external stimuli and vigilance states. A possible alternative interpretation is that individuals who engage in more frequent daydreaming may pay greater attention to their internal states and subjective experiences. Such a heightened introspective awareness might in turn facilitate the encoding and recall of dream experiences.

Our analyses revealed an important role of overnight sleep patterns as extracted from actigraphic data. Specifically, we found that individuals who typically have long sleep episodes coupled with a low proportion of deep N3 sleep, exhibit a heightened probability of dream recall compared to those experiencing shorter, N3-rich sleep. This observation aligns with prior investigations describing a positive correlation between sleep duration and dream recall[61] and a negative relationship between the occurrence of slow waves typical of NREM sleep and dreaming[62]. Indeed, not only the probability of reporting a dream upon awakening decrease in parallel with the increase in slow wave activity during the deepening of NREM sleep, from N1 to N3[14], but, even within the same sleep stage (be that NREM or REM), higher slow wave activity is more often observed when individuals report no dream experiences[41,62–64]. Sleep slow waves are primarily local events that result from an oscillation of cortical neuronal populations between a depolarized, active state, and a hyperpolarized, silent state. Their asynchronous occurrence across cortical areas is thought to cause a breakdown in cortical connectivity and impair information integration, a key prerequisite for the emergence of consciousness[65,66]. Importantly, slow waves are homeostatically regulated, so that they increase in number and amplitude after sleep deprivation/restriction and decrease across sleep cycles, during a night of sleep (or a nap[67]). In this light, a large amount of N3 sleep may indicate a high sleep pressure, with a strong slow wave activity that may decrease the probability of experiencing dreams regardless of the sleep stage from which the sleeper wakes up.

In addition, our results demonstrate that dream recall frequency is characterized by seasonal fluctuations, being lower during Winter as compared to Spring and Autumn. While our actigraphic data did not allow us to detect macrostructural sleep changes that could explain the variations in dream recall, we were not able to investigate possible finer variations in sleep macro- or micro-structure. Such variations could contribute to determining the decrease in dream recall observed in Winter and should be the object of further investigations.

Next, we investigated the individual factors that may affect the retrieval of dream content and lead to the perception of having been dreaming without managing to recall any feature of that experience, so-called *white dreams*[32]. Notably, we observed that the same sleep patterns associated with morning dream recall—namely, long, light sleep bouts—are equally associated with recalling dream content and forgetting it. This observation

suggests that white dreams and contentful dreams both reflect true, generated dreams with different degrees of recall of specific aspects of the experience. Interestingly, in line with previous work, we did not find potential associations between white dreams and visual or verbal memory[32], suggesting that memory processes regarding dream content may not be affected by general memory skills. Instead, we found that individuals with a higher vulnerability to interference tend to more often forget the content of their dreams upon awakening. This observation is consistent with previous evidence suggesting that the memory of a dream may be lost if interference occurs between dream experience and retrieval[68]. Indeed, higher resilience to interference may allow individuals to maintain the focus of attention and memory on the dream in spite of situational (e.g., turning off the alarm sound or talking with the bed partner) or internal (e.g., thoughts about the upcoming schedule or thinking about current concerns) interferences. This observation provides support to the so-called interference hypothesis for dream recall, which postulates that the dream memory trace persists as long as there is no distraction or interference[68,69]. According to this idea, dream recall is more likely if the dreamer pays attention to and is able to maintain their focus on the dream experience immediately after the awakening. This idea may also explain the effects of attitude towards dreaming on dream recall, as described by the so-called lifestyle hypothesis. Indeed, individuals with a higher interest in dreams may be expected to put more attention on their own dream experiences upon awakening[70].

Previous work suggested that dream recall may decrease with age[18,51] and that female individuals may recall more dreams than male individuals[17,51,71]. Neither of these findings was confirmed here. We hypothesize that such effects might be actually mediated by other variables. Indeed, for instance, we found that the attitude towards dreaming, positively associated with dream recall, is higher in female than in male participants[34]. Moreover, our results suggest that aging may be associated with changes in sleep patterns—and in particular with a decrease in the long, light sleep bouts—that may in turn affect dream generation processes. However, we found an independent effect of age on content recall, so that aging is associated with a higher probability of reporting *white dreams*. The mechanism underlying this association is unclear, and such an effect may actually be mediated by variations in other cognitive processes not investigated here, such as working memory skills (see ref. 72).

## Limitations

While the dream recall rate reported in this study (CD = 4.1 per week; CD + WD = 5.0 per week) may appear relatively high compared to previous research, commonly pointing to a range of 1–3 recalled dreams per week (e.g., ref. 51; but see also ref. 40), the observed values may be explained by the methodological approach we adopted. For example, while here we instructed participants to consider as dreams any subjective conscious experience occurring during sleep (e.g., ref. 41), other studies allowed

participants to apply their own definition of what constitutes a dream, or limitedly focused on perceptual, vivid, and rich (REM-like) experiences. Moreover, while most studies relied on written morning reports, here we used oral reporting. Importantly, written reports are simpler and faster to process for experimenters but entail a greater effort from participants and might be expected to also more easily determine interference effects[42]. Since these and other methodological choices may affect dream recall estimates, we argue that the field would benefit from consensus-level standardization and improvements in methodological reporting to enable more reliable comparisons across studies and further research advancement.

In this work, objective information about sleep patterns was obtained using actigraphic devices and a portable electrophysiological recording system (the DREEM headband). While these instruments may offer valuable information about sleep duration, quality, and structure, they are known to have lower reliability with respect to standard laboratory polysomnography. At the same time, they currently represent the best viable compromise for longitudinal studies aimed at collecting data in naturalistic conditions and in large samples. In the context of this study, both manual and automatic approaches were used to identify and exclude recordings related to clear device removal or malfunctioning. Moreover, cross-comparisons across actigraphy and DREEM recordings showed a high level of consistency, providing further support to the validity of estimates derived from these devices (see Supplementary Figs. 3–9).

## Conclusions
Our study demonstrates that specific inter-individual (trait) and intra-individual (state) variables influence the likelihood of having and recalling a dream experience. Notably, our findings show that similar overnight sleep patterns increase the probability of both contentful and *white dreams*, and that the memory retention for dream content may be primarily lost due to interference by external or internal factors. These observations support the notion that *white dreams* represent actual dream experiences, with memories of their content fading upon waking.

## Data availability
All data for this manuscript are freely available on the Open Science Framework at: https://osf.io/3qwsm/.

## Code availability
All code for this manuscript is freely available on the Open Science Framework at: https://osf.io/3qwsm/.

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

## Acknowledgements

This work was supported by a grant from the BIAL Foundation (#091/2020; to G.Be., M.B., and V.E.). The funders had no role in the conceptualization, design, data collection, analysis, decision to publish, or preparation of the manuscript. The authors thank Elena Capriglia, Monica Di Giuliano, Francesco Lomi, and Aurora Salina for their help with data collection and preprocessing.

## Author contributions

Conceptualization: G.Be., M.B., and V.E.; investigation: V.E., G.Bo., and B.P.; methodology: G.H., G.Be., and V.E.; software: G.H., G.Be., V.E., and D.B.; formal analysis: V.E., D.B., G.Be., and G.H.; visualization: V.E. and G.B.; supervision: G.H. and G.B.; funding acquisition: G.Be. and M.B.; project administration: G.Be; data curation: V.E.; writing—original draft: V.E. and G.Be.; writing—review and editing: All authors. All authors have read and agreed to the published version of the manuscript.

## Competing interests

The authors declare no competing interests.
