## [Transparent Peer Review file · Communications Psychology]

The individual determinants of morning dream recall

Corresponding Author: Professor Giulio Bernardi

Version 0:

Decision Letter:

Dear Professor Bernardi,

Thank you for your patience during the peer-review process. Your manuscript titled "The individual determinants of morning dream recall" has now been seen by 2 reviewers, and I include their comments at the end of this message. They find your work of interest but raised some important points. We are interested in the possibility of publishing your study in Communications Psychology, but would like to consider your responses to these concerns and assess a revised manuscript before we make a final decision on publication.

We therefore invite you to revise and resubmit your manuscript, along with a point-by-point response to the reviewers. Please highlight all changes in the manuscript text file.

Editorially, we consider the following issues key:

First, please provide the additional analyses that are needed to alleviate the referees' concerns about the methodological soundness of the approach and ambiguities in the interpretation.

Second, please ensure that all caveats are clearly discussed (including in a dedicated "Limitations" section in the manuscript) and that the relevant literature is presented fairly and comprehensively.

Third, given the absence of clearly stated hypotheses in the Introduction and the absence of a preregistration link, we editorially assume that the work was explorative rather than confirmatory in nature. If this is the case, please refrain from formulating hypotheses in the Introduction (it is however permissible to discuss what framework your results may tentatively support or contradict in the Discussion). In either case, as per our guidelines in the attached checklist, please state clearly whether any of the hypotheses or the analysis plan were or were not preregistered.

I am attaching an Editorial Requests Table that details critical reporting requirements for the revised manuscript. Please attend to each item and ensure your manuscript is fully compliant. We are requesting that your manuscript aligns with these requirements as this facilitates the evaluation of your manuscript, reducing delays in re-review and potential future acceptance. If your revised manuscript is not aligned with these requests on major issues, such as those concerning statistics, it may be returned to you for further revisions without re-review. Additional information can be found in our style and formatting guide Communications Psychology formatting guide.

Please use the following link to submit your

- revised manuscript,
- point-by-point response to the referees' comments,
- cover letter (as a separate document),
- the Editorial Policy Checklist (see below),
- the Reporting Summary (see below), and
- the completed Editorial Request Table (attached):

Link Redacted

Best regards,

Xiaoqing Hu

Xiaoqing Hu, PhD
Editorial Board Member
Communications Psychology
orcid.org/0000-0001-8112-9700

REVIEWER EXPERTISE:

Reviewer #1: dream recall, actigraphy, individual differences

Reviewer #2: sleep, actigraphy, statistics

REVIEWER REPORTS:

Reviewer #1 (Remarks to the Author):

Overall, the study elicited interesting data, but the theoretical background and the presentation of the data should be more focused. I raised a few topics and suggest a thoroughly carried out re-writing of the paper.

1. In the introduction, there are many sentences without adequate references, e.g., "Dream experiences draw on previously acquired memories and beliefs and, thus, present relevant aspects of continuity with thoughts, concerns, and salient experiences of our waking self." Or "In light of this, they are believed to represent an important window on—and to potentially have a direct role in— sleep-dependent processes involving learning and memory consolidation." Or "Moreover, dreams have a tight relationship with psychophysical health." Or "Finally, the study of dreaming and dreamless sleep is regarded as a fundamental experimental model in the search for the functional bases of human consciousness." Or "Previous studies suggested that factors such as a positive attitude towards dreaming, frequent daydreaming, a high level of anxiety, female gender, and young age may be associated with higher dream recall frequency." Or "This picture is further complicated by the inherent foundation of dream studies, to some degree, on the assumption that reports provided by individuals upon awakening are a reliable reflection of dream occurrence and content."

The authors address relevant issues but fail to reference the empirical work already done in the field.

2. The authors claim that "Prospective studies conducted so far are sparse and hampered by significant limitations." But they didn't cite any of those studies, e.g. Schredl, M., & Basak, M. (2020). A diary study of dream recall: Successful dream recall and contentless dreams. *International Journal of Dream Research*, 13(1), 123-126.

<https://doi.org/10.11588/ijodr.2020.1.70982>. Just to name one study.

3. In addition to the lack of a comprehensive overview, let's say about the already existing research about factors related to dream recall (e.g., cognitive functioning/memory and dream recall), the authors did not present any theoretical model for dream recall in the introduction, e.g., one finding fits in the interference hypothesis. Therefore, the last section of the introduction lacks clearly stated hypotheses.

4. The authors emphasize the difference between retrospective measures of dream recall and prospective measures. As there is a literature on this topic, just to name a few: Aspy, D. J., Delfabbro, P., & Proeve, M. (2015). Is dream recall underestimated by retrospective measures and enhanced by keeping a logbook? A review. *Consciousness and Cognition*, 33, 364-374. <https://doi.org/10.1016/j.concog.2015.02.005> or Zunker, M., Althoff, H. K., Apel, J., Lässig, H. S., Schültke, L., & Schredl, M. (2015). Comparing questionnaire and diary measures for eliciting nightmare frequency. *International Journal of Dream Research*, 8, 129-134. <https://doi.org/10.11588/ijodr.2015.2.23395>. This research should be carefully reviewed by the authors, as there are pros and cons to both approaches, it is not the case that prospective is better (increase of dream recall by keeping a diary is biasing the results).

5. Figure 1c is not telling much, I cannot differentiate between $r = 0.1$ or $r = 0.3$, one might be significant, the other not. Please report the zero-order correlations explicitly in the supplement.

6. I didn't understand fully the procedure regarding measuring dream recall. What about awakenings during the night, where participants allowed to record whether they woke up with dream recall, or was dream recall only measured in the morning. This is a serious bias and should be discussed.

7. "On average, CD+WD probability was 0.72 ± 0.22 , corresponding to 5.04 ± 1.54 dreams per week, in line with previous

observations concerning dream recall frequency [19].” The study cited includes data showing that successful dream recall (CD) is about once per week in a representative sample using retrospective measures. Thus, the statement “in line” is incorrect. It is the contrary, the authors have recruited a sample of high dream recallers and, thus, have not found gender and age effects like reported in large-scaled studies (e.g., Schredl, M., Berres, S., Klingauf, A., Schellhaas, S., & Göritz, A. S. (2014). The Mannheim Dream questionnaire (MADRE): Retest reliability, age and gender effects. *International Journal of Dream Research*, 7, 141-147. <https://doi.org/10.11588/ijodr.2014.2.16675>” and meta-analysis (Schredl, M., & Reinhard, I. (2008). Gender differences in dream recall: A meta-analysis. *Journal of Sleep Research*, 17, 125-131. <https://doi.org/10.1111/j.1365-2869.2008.00626.x>). This clearly limits the generalizability of the study and should be discussed.

8. I have problems with the interpretation of PC2, in my opinion, the main factor is sleep duration, the negative correlation does make sense as longer sleep includes a smaller percentage of N3 sleep (as this sleep stage is lacking in the second part of the night). So the authors should check on studies, looking at the association between sleep duration and dream recall, e.g., Schredl, M., & Fulda, S. (2005). Dream recall and sleep duration: state or trait factor. *Perceptual and Motor Skills*, 101, 613-616. Or Schredl, M., & Reinhard, I. (2008). Dream recall, dream length and sleep duration: state or trait factor. *Perceptual and Motor Skills*, 106, 633-636. These findings are somewhat in line with the data presented.

9. In the section “Seasonal variations”, I do not see any Ns, how many participants are in the different group. And I also do not understand why sleep parameters were not controlled for in the analysis of possible seasonal effects on dream recall.

10. Lastly, the authors have also elicited dream recall frequency (retrospectively). Based on their introduction, it would be very interesting to test whether influencing factors (the whole range) correlate differently for the two measures, and how high the correlation between the two measures are. This might be lower compared to other samples because of the high recaller bias in this study.

Reviewer #2 (Remarks to the Author):

The authors describe a large longitudinal study on dream reports. Dream recall was measured at awakening in the morning and sleep was assessed using actigraphy. A subset of participants also used a dry electrode polysomnography device, which was analysed using an automated sleep scoring algorithm. Inter-individual differences were assessed and related to the dream recall and the authors find that attitudes towards dreaming, proneness to mind wandering and long light sleep (a principal component derived from the actigraphy data) were positively related to increased dream recall. Contentful dream recall vs. white dream recall (recall of the feeling of dreaming without knowing the content) was associated with age and vulnerability to interference. Of note, none of the memory measures was related to dream recall. In addition, the authors investigated the relationship between dream recall and seasons. Here they found that dream recall is lower in winter than in spring. Overall the study is interesting and conducted at a high level. I only have a few concerns that should be addressed.

1. From the methods, it was unclear to me, if the PCA took into account that some of the measurement came from the same individuals, i.e., were dependent. This seems like an important issue of the data structure that would bias the PCA.

2. I am aware of one study showing acceptable performance of the DREEM headband, however, there are considerable reservations in the field of sleep research regarding its validity as a polysomnography device. This should be added as a limitation. In the supplement, the authors should provide example data for good, medium and poor quality data in their sample. In addition, I would like to see the performance of the DREEM algorithm versus another automated scoring (i.e., USLEEP, Perselev et al, 2021). Ideally, the authors also show scoring of a subset of the data by experienced technicians.

3. As far as I understand the relationships between actigraphy and DREEM data were made on the aggregate level, i.e., values for entire nights of sleep. To check the validity of these relationships the authors should compare the time series of the actigraphy data to the DREEM data.

4. I was somewhat surprised about the frequency of contentful dreams. Although the authors report this to be in line with previous research, I wonder how representative the sample is. How did the authors recruit participants and is it possible that they had an increased interest in dreams at the outset. If this is the case, how does this limit the research?

5. In general the figures are informative, but quite hard to read. The asterisks are too small and overall the size of graphical elements could be optimised. This is especially the case for Figure 1c.

EDITORIAL POLICIES

We ask that you ensure your manuscript complies with our editorial policies and reporting requirements.

To that end, we require revised manuscripts to be accompanied by two completed items: a reporting summary that collects information on study design and procedure, and an editorial policy checklist that verifies compliance with all required editorial policies.

- <https://www.nature.com/documents/nr-reporting-summary.zip>>Nature Research Reporting Summary
- <https://www.nature.com/documents/nr-editorial-policy-checklist.pdf>>Editorial Policy Checklist

All points on the policy checklist must be addressed. Your revised manuscript can only be sent back to the referees if these checklists are completed and uploaded with the revision.

Notes: If you have submitted a Stage 1 Registered Report, Review, Primer, Comment, or Perspective you do not need to submit these forms. If you have already submitted these forms, you may disregard this request.

Communications Psychology is committed to improving transparency in authorship. As part of our efforts in this direction, we are now requesting that all authors identified as 'corresponding author' create and link their Open Researcher and Contributor Identifier (ORCID) with their account on the Manuscript Tracking System prior to acceptance. ORCID helps the scientific community achieve unambiguous attribution of all scholarly contributions. You can create and link your ORCID from the home page of the Manuscript Tracking System by clicking on 'Modify my Springer Nature account' and following the instructions in the link below. Please also inform all co-authors that they can add their ORCID to their accounts and that they must do so prior to acceptance.

Version 1:

Decision Letter:

Dear Professor Bernardi,

Your manuscript titled "The individual determinants of morning dream recall" has now been seen by our reviewers, whose comments appear below.

In light of their advice, we would in principle - provided Reviewer #2's requests are met in a final revision - be happy to publish this suitably revised version in Communications Psychology.

We therefore invite you to revise your paper one last time to address the remaining concerns of our reviewer and a list of editorial requests. At the same time we ask that you edit your manuscript to comply with our format requirements and to maximise the accessibility and therefore the impact of your work.

EDITORIAL REQUESTS:

SUBMISSION INFORMATION:

OPEN ACCESS:

* TRANSPARENT PEER REVIEW: Communications Psychology uses a transparent peer review system. On author request, confidential information and data can be removed from the published reviewer reports and rebuttal letters prior to publication. If you are concerned about the release of confidential data, please let us know specifically what information you would like to have removed. Please note that we cannot incorporate redactions for any other reasons.

Link Redacted

Best regards,

Marike Schiffer & Xiaoqing Hu

Marike Schiffer, PhD
Chief Editor
Communications Psychology

REVIEWERS' COMMENTS:

Reviewer #1 (Remarks to the Author):

The changes have improved - in my view - the paper.

Reviewer #2 (Remarks to the Author):

The authors have substantially improved the manuscript. However, there remain two points. The first is just to give them credit for actually finding a good way to deal with structured data and should strengthen the ms. without a lot of work. The second is a crucial point that should have been addressed in the first revision.

Reviewer #2 my original point 1: This is convincing. Please add a sentence to the manuscript that this was done and report the correlations between MFA and PCA.

Reviewer #2 my original point 2: I have close to 20 years of experience in sleep research. In this time I have learned enough about sleep-EEG that seeing some raw data will be informative irrespective the derivation. This is also the case for many of our colleagues. Therefore, I ask the authors to provide this in the supplement. USLEEP is able to handle a broad variety of derivations and it would be informative to see how it performs, even if the ground truth remains elusive. I do not have confidence in the DREEM data unless this is added to the manuscript.

We are extremely grateful to the Editor and the anonymous Reviewers for their careful assessment of our work and for their valuable comments and suggestions, which helped us improve our manuscript's quality. Please find below our point-by-point response to each comment and a description of all changes made to the manuscript. For the Editor's and Reviewers' convenience, we reported in our responses the portions of the manuscript that we revised, highlighting changed text in red color.

Reviewer #1

Overall, the study elicited interesting data, but the theoretical background and the presentation of the data should be more focused. I raised a few topics and suggest a thoroughly carried out re-writing of the paper.

We thank the Reviewer for their careful assessment of our work and their insightful comments.

1. In the introduction, there are many sentences without adequate references, e.g., “Dream experiences draw on previously acquired memories and beliefs and, thus, present relevant aspects of continuity with thoughts, concerns, and salient experiences of our waking self.” Or “In light of this, they are believed to represent an important window on —and to potentially have a direct role in— sleep-dependent processes involving learning and memory consolidation.” Or “Moreover, dreams have a tight relationship with psychophysical health.” Or “Finally, the study of dreaming and dreamless sleep is regarded as a fundamental experimental model in the search for the functional bases of human consciousness.” Or “Previous studies suggested that factors such as a positive attitude towards dreaming, frequent daydreaming, a high level of anxiety, female gender, and young age may be associated with higher dream recall frequency.” Or “This picture is further complicated by the inherent foundation of dream studies, to some degree, on the assumption that reports provided by individuals upon awakening are a reliable reflection of dream occurrence and content.” The authors address relevant issues but fail to reference the empirical work already done in the field.

We thank the Reviewer for having pointed our attention toward this issue. Several citations were added to the Introduction section, as detailed below. Moreover, we expanded the paragraph related to the individual variables associated with a higher dream recall frequency when discussing the theoretical framework, as suggested by the reviewer in the comment #3 below.

Page 3, Line 40: “Dream experiences draw on previously acquired memories and beliefs and, thus, present relevant aspects of continuity with thoughts, concerns, and salient experiences of our waking self [REF1, REF2]. In light of this, they are believed to represent an important window on —and to potentially have a direct role in— sleep-dependent processes involving learning and memory consolidation [REF3, REF4]. Moreover, dreams have a tight relationship with psychophysical health [REF5].”

REF1: Domhoff, G. W. (2010). Dream content is continuous with waking thought, based on preoccupations, concerns, and interests. *Sleep Medicine Clinics*, 5, 203-215.

REF2: Malinowski, J. E., & Horton, C. L. (2021). Dreams reflect nocturnal cognitive processes: Early-night dreams are more continuous with waking life, and late-night dreams are more emotional and hyperassociative. *Consciousness and Cognition*, 88, 103071.

REF3: Bloxham, A., & Horton, C. L. (2024). Enhancing and advancing the understanding and study of dreaming and memory consolidation: Reflections, challenges, theoretical clarity, and methodological considerations. *Consciousness and Cognition*, 123, 103719.

REF4: Hudachek, L., & Wamsley, E. J. (2023). A meta-analysis of the relation between dream content and memory consolidation. *Sleep*, 46(12), zsad111.

REF5: Siclari, F., Valli, K., & Arnulf, I. (2020). Dreams and nightmares in healthy adults and in patients with sleep and neurological disorders. *The Lancet Neurology*, 19(10), 849-859.

Page 3, Line 47: “Finally, the study of dreaming and dreamless sleep is regarded as a fundamental experimental model in the search for the functional bases of human consciousness [REF6, REF7]. Indeed, as compared to task-based protocols exploring wakefulness conscious experiences, the study of dreams is naturally less influenced by confounding effects such as changes in attention, stimulus and task processing, task performance, and response preparation”.

REF6: Koch, C., Massimini, M., Boly, M., Tononi, G. (2016). Neural correlates of consciousness: progress and problems. *Nat Rev Neurosci*; 17(5):307-21

REF7: Tononi, G., Boly, M., & Cirelli, C. (2024). Consciousness and sleep. *Neuron*, 112(10), 1568-1594.

Page 4, Line 86: “This picture is further complicated by the inherent foundation of dream studies, to some degree, on the assumption that reports provided by individuals upon awakening are a reliable reflection of dream occurrence and content [REF8]”.

REF8: Windt, J. M. (2013). Reporting dream experience: Why (not) to be skeptical about dream reports. *Frontiers in human neuroscience*, 7, 708.

2. The authors claim that “Prospective studies conducted so far are sparse and hampered by significant limitations.” But they didn’t cite any of those studies, e.g. Schredl, M., & Basak, M. (2020). A diary study of dream recall: Successful dream recall and contentless dreams. International Journal of Dream Research, 13(1), 123-126. <https://doi.org/10.11588/ijodr.2020.1.70982>. Just to name one study.

We added references in support of the sentence indicated by the Reviewer. Please note that we cited a review article including extensive discussions concerning issues and limitations affecting the study of dream recall, as these may offer readers a more complete picture than a few representative examples. Moreover, we included here the references suggested by the reviewer in comment #4.

Page 3, Line 73: *Indeed, the available evidence is mostly based on retrospective measures potentially affected by biases such as memory- and personality-related distortions. Prospective studies conducted so far are sparse and hampered by significant limitations as, due to their higher costs, these investigations were typically performed on relatively small samples and explored only one or few variables potentially affecting dream recall (see [REF9, REF10, REF11] for a discussion about pros and cons of prospective and retrospective approaches)*”.

REF9: Schredl, M. (1999). Dream recall: research, clinical implications and future directions. *Sleep and Hypnosis, 1*(2), 72-81.

REF10: Aspy, D. J., Delfabbro, P., & Proeve, M. (2015). Is dream recall underestimated by retrospective measures and enhanced by keeping a logbook? A review. *Consciousness and Cognition, 33*, 364-374.

REF11: Zunker, M., Althoff, H. K., Apel, J., Lässig, H. S., Schültke, L., & Schredl, M. (2015). Comparing questionnaire and diary measures for eliciting nightmare frequency. *International Journal of Dream Research, 8*, 129-134.

3. In addition to the lack of a comprehensive overview, let’s say about the already existing research about factors related to dream recall (e.g., cognitive functioning/memory and dream recall), the authors did not present any theoretical model for dream recall in the introduction, e.g., one finding fits in the interference hypothesis. Therefore, the last section of the introduction lacks clearly stated hypotheses.

We thank the Reviewer for having pointed our attention to these issues. A new text section was added in the Introduction to offer the readers a brief overview concerning previous investigations about the determinants of dream recall frequency (see below). Please note that, given the wide heterogeneity of methodologies and results and the consequent difficulty of their interpretation, we chose to highlight only some of the most relevant findings, and directed the interested reader to more thorough literature reviews.

Page 3, Line 65: *“While the sleep stage preceding the awakening is considered a key determinant for whether or not a dream will be reported, evidence indicates that dream recall probability fluctuates greatly both within and across individuals. Such a variability attracted public and scientific attention during the recent pandemic, when an abrupt surge in morning dream recall was reported worldwide. Yet, our current understanding of the factors influencing dream generation and recall is scarce and lacks a univocal picture. For instance, while several studies found female sex [REF12], younger age [REF13], a positive attitude towards dreaming, frequent daydreaming, and fantasy proneness, as consistently associated with a higher dream recall frequency [REF14, REF15, REF16], other investigations produced partially inconsistent or contradictory results (e.g., [REF17]). Results concerning the possible involvement of other personality or*

cognitive factors, such as visual and verbal memory, produced even more inconsistent results with some studies indicating a positive association [REF19, REF19] and others observing no significant predictive power [REF20, REF21]. These inconsistencies could be explained by differences in employed definitions and applied experimental approaches across studies (for detailed reviews see [REF22, REF23]). Indeed, the available evidence is mostly based on retrospective measures potentially affected by biases such as memory- and personality-related distortions. Prospective studies conducted so far are sparse and hampered by significant limitations as, due to their higher costs, these investigations were typically performed on relatively small samples and explored only one or few variables potentially affecting dream recall [...]”.

REF12: Schredl, M. (2010). Explaining the gender difference in dream recall frequency. *Dreaming*, 20(2), 96–106.

REF13: Nielsen, T. (2012). Variations in dream recall frequency and dream theme diversity by age and sex. *Frontiers in neurology*, 3, 106.

REF14: Watson, D. (2003). To dream, perchance to remember: Individual differences in dream recall. *Personality and individual differences*, 34(7), 1271-1286.

REF15: Giesbrecht, T., & Merckelbach, H. (2006). Dreaming to reduce fantasy?—Fantasy proneness, dissociation, and subjective sleep experiences. *Personality and Individual Differences*, 41(4), 697-706.

REF16: Schredl, M., Wittmann, L., Ciric, P., & Götz, S. (2003). Factors of home dream recall: a structural equation model. *Journal of Sleep Research*, 12(2), 133-141.

REF17: Beaulieu-Prévost, D., & Zadra, A. (2005). Dream recall frequency and attitude towards dreams: A reinterpretation of the relation. *Personality and Individual Differences*, 38(4), 919-927.

REF18: Blagrove, M., & Pace-Schott, E. F. (2010). Trait and neurobiological correlates of individual differences in dream recall and dream content. *International Review of Neurobiology*, 92, 155-180.

REF19: Butler, S. F., & Watson, R. (1985). Individual differences in memory for dreams: The role of cognitive skills. *Perceptual and Motor Skills*, 61(3), 823-828.

REF20: Eichenlaub, J. B., Nicolas, A., Daltrozzo, J., Redouté, J., Costes, N., & Ruby, P. (2014). Resting brain activity varies with dream recall frequency between subjects. *Neuropsychopharmacology*, 39(7), 1594-1602.

REF21: Cohen, D. B. (1971). Dream recall and short-term memory. *Perceptual and motor skills*, 33(3), 867-871.

REF22: Schredl, M. (2018). *Researching dreams: The fundamentals*. Springer.

REF23: Nemeth, G. (2023). The route to recall a dream: Theoretical considerations and methodological implications. *Psychological Research*, 87(4), 964-987.

For what concerns the hypotheses and theoretical framework of the study, we note that, as also highlighted by the Editor, the present work was exploratory in nature and not aimed at testing a specific hypothesis regarding the mechanisms of dream recall. As per specific Editor's request, we avoided indicating a defined hypothesis or theoretical framework in the Introduction section. However, we expanded the Discussion section to better clarify how our results relate to existing theories about dream recall, and in particular the so-called *interference hypothesis*.

Page 14, Line 483: *“Instead, we found that individuals with a higher vulnerability to interference tend to more often forget the content of their dreams upon awakening. This observation is consistent with previous evidence suggesting that the memory of a dream may be lost if interference occurs between dream experience and retrieval [38]. Indeed, higher resilience to interference may allow individuals to maintain the focus of attention and memory on the dream in spite of situational (e.g., turning off the alarm sound or talking with the bed partner) or internal (e.g., thoughts about the upcoming schedule or thinking about current concerns) interferences. This observation provides support to the so-called interference hypothesis for dream recall, which postulates that the dream memory trace persists as long as there is no distraction or interference ([REF24], [REF25]). According to this idea, dream recall is more likely if the dreamer pays attention to and is able to maintain their focus on the dream experience immediately after the awakening. This idea may also explain the effects of attitude towards dreaming on dream recall, as described by the so-called life-style hypothesis. Indeed, individuals with a higher interest in dreams may be expected to put more attention on their own dream experiences upon awakening [REF26].”*

REF24: Cohen, D. B., & Wolfe, G. (1973). Dream recall and repression: Evidence for an alternative hypothesis. *Journal of Consulting and Clinical Psychology*, 41(3), 349.

REF25: Cohen, D. B., & MacNeilage, P. F. (1974). A test of the salience hypothesis of dream recall. *Journal of consulting and clinical psychology*, 42(5), 699.

REF26: Schredl, M., & Montasser, A. (1996). Dream recall: State or trait variable? Part I: Model, theories, methodology and trait factors. *Imagination, Cognition and Personality*, 16(2), 181-210.

4. The authors emphasize the difference between retrospective measures of dream recall and prospective measures. As there is a literature on this topic, just to name a few: Aspy, D. J., Delfabbro, P., & Proeve, M. (2015). Is dream recall underestimated by retrospective measures and enhanced by keeping a logbook? A review. *Consciousness and Cognition*, 33, 364-374. <https://doi.org/10.1016/j.concog.2015.02.005> or Zunker, M., Althoff, H. K., Apel, J., Lässig, H. S., Schültke, L., & Schredl, M. (2015). Comparing questionnaire and diary measures for eliciting nightmare frequency. *International Journal of Dream Research*, 8, 129-134. <https://doi.org/10.11588/ijodr.2015.2.23395>. This research should be carefully reviewed by the authors, as there are pros and cons to both approaches, it is not the case that prospective is better (increase of dream recall by keeping a diary is biasing the results).

We agree with the Reviewer that prospective and retrospective measures have both pros and cons. We modified the main text to direct the interested readers toward the works indicated by the Reviewer, as reported in our response to comment #2.

Nevertheless, issues such as the one mentioned by the Reviewer may be partially accounted for when using prospective measures, as relative changes in dream recall across experimental days may be modeled (as done in the present study). Prospective approaches also allow to evaluate and account for day-by-day changes in recall as they do not have to necessarily rely on summary statistics (see our response to comment #10). In this light, they offer greater power and more statistical flexibility than retrospective approaches.

5. Figure 1c is not telling much, I cannot differentiate between $r = 0.1$ or $r = 0.3$, one might be significant, the other not. Please report the zero-order correlations explicitly in the supplement.

The correlations included in Figure 1c were tested to evaluate whether the selected predictors of dream recall had any strong associations, as this could have affected our statistical approaches (of note, no such strong correlations emerged).

To accommodate the Reviewer's request, we added to Figure 1c markers indicating statistically significant correlations ($p < 0.05$, FDR correction; see below, Fig. R1). To better illustrate relative differences in the strength of differences we also modified the color scale by changing the represented range from $[-1, +1]$ to $[-0.5, +0.5]$. An additional matrix showing the actual correlation values within each cell was added as Supplementary Fig. S1 (see below, Fig. R2). Moreover, we clarified the rationale for the inclusion of Figure 1c in the Results section, as follows.

Page 10, Line 320: *“The following predictors were selected based on previous literature: age, sex, years of education, attitude towards dreaming, vulnerability to cognitive interference, verbal memory, visuospatial memory, trait anxiety, vividness of visual imagery, proneness to mind-wandering, self-reported sleep quality, and self-reported chronotype (relative correlations among these predictors are shown in Fig. 1c)”*.

Figure R1. Correlation (Spearman's correlation coefficient) between demographic, psychological and cognitive variables derived from questionnaires and tests. *Black dots indicate significant associations ($q < 0.05$, FDR correction). A moderate significant correlation was found between age and vulnerability to interference ($r = -0.46$). Relatively small but significant correlations also emerged between trait anxiety and subjective sleep quality ($r = 0.28$), proneness to mind wandering ($r = 0.20$), vividness of visual imagery ($r = -0.21$), chronotype ($r = 0.20$), and vulnerability to interference ($r = 0.19$), between attitude towards dreaming and education ($r = -0.24$), between chronotype and age ($r = 0.26$), between chronotype and vulnerability to interference ($r = -0.21$), as well as between verbal memory and sex ($r = -0.19$). Also see Supplementary Fig. S1.*

Figure R2. Correlation (Spearman's correlation coefficient) between demographic, psychological and cognitive variables derived from questionnaires and tests. This plot is the same depicted in Fig. 1c, but here correlation coefficients are shown in each cell. White text indicates significant effects ($q < 0.05$, FDR correction).

6. I didn't understand fully the procedure regarding measuring dream recall. What about awakenings during the night, where participants allowed to record whether they woke up with dream recall, or was dream recall only measured in the morning. This is a serious bias and should be discussed.

We apologize for the lack of clarity regarding the procedure. All participants received standardized and detailed instructions regarding which experiences they had to consider in their verbal reports (please also see our response to comment #7). Specifically, they were instructed to report only the last dream experience that they had in the morning just before the awakening, following the same approach commonly used for serial awakening studies (e.g., Siclari et al., *Frontiers in Psychology*, 2013). This choice was made to minimize potential biases related to the occurrence of nightly awakenings determined by external and potentially random causes (e.g., discomfort due to high or low temperature, presence of bed partners or domestic animals, etc.). Of note, awakenings during the night may be associated with a relatively strong sleep inertia due to higher homeostatic and circadian sleep pressure. Such a condition may affect memory encoding and retrieval - especially if retrieval is attempted in the morning. Of great importance is also the fact that sleep pressure and sleep inertia may negatively affect the amount of motivation of participants, who may choose to not report the experiences they had during the night in order to rapidly return to sleep (when reports are collected after each awakening).

We are aware that some previous investigations asked participants to report all dreams that they could recall regardless of when they occurred throughout the night. To eliminate potential ambiguities, we modified the manuscript to better describe the instructions given to participants (see comment #7).

7. "On average, CD+WD probability was 0.72 ± 0.22 , corresponding to 5.04 ± 1.54 dreams per week, in line with previous observations concerning dream recall frequency [19]." The study cited includes data showing that successful dream recall (CD) is about once per week in a representative sample using retrospective measures. Thus, the statement "in line" is incorrect. It is the contrary, the authors have recruited a sample of high dream recallers and, thus, have not found gender and age effects like reported in large-scaled studies (e.g., Schredl, M., Berres, S., Klingauf, A., Schellhaas, S., & Göritz, A. S. (2014). The Mannheim Dream questionnaire (MADRE): Retest reliability, age and gender effects. *International Journal of Dream Research*, 7, 141-147. <https://doi.org/10.11588/ijodr.2014.2.16675”>; and meta-analysis (Schredl, M., & Reinhard, I. (2008). Gender differences in dream recall: A meta-analysis. *Journal of Sleep Research*, 17, 125-131. <https://doi.org/10.1111/j.1365-2869.2008.00626.x>). This clearly limits the generalizability of the study and should be discussed.

We agree with the Reviewer that the cited study (Schredl, *Perceptual and Motor Skills*, 2008) does not support the consistency of our recall rate with those of previous investigations. Indeed, the reference was erroneously included in place of another reference from the same author (Schredl et al., *Dreaming*, 2002). This latter work used both a questionnaire and a dream diary to estimate dream recall frequency (measured as the number of mornings with recall relative to total number of experimental days) in 285 individuals (212 females). The dream diary assessed the occurrence of both contentful (CD) and white dreams (WD).

The author found an overall report rate of 2.0 ± 1.4 CDs per week, and estimated from the retrospective questionnaire a similar recall rate, corresponding to 2.1 ± 0.7 dreams per week. This value is not much smaller than the one we obtained from our questionnaire (representing the Italian translation of the one used in Schredl's work), corresponding to 2.7 dreams per week. When combining CDs and WDs, Schredl obtained a recall rate of 4.7 ± 1.7 dreams per week; again a value not much smaller than the one we reported in our study (5.0 ± 1.5 dreams per week).

More in general, we note that dream recall rates vary greatly in the literature depending on the adopted method of investigation and population under scrutiny. For instance, recall rates have been reported to be 3 to 10 times higher for diaries or checklist logbooks relative to retrospective questionnaires (Nemeth, *Psychological Research*, 2023). Moreover, some studies counted separately dream recall instances within the same night (e.g., in the case of multiple overnight awakenings), while others only focused on whether or not at least one dream was recalled in the morning. For example, the work by Zadra and Robert (Zadra and Robert, *Consciousness and Cognition*, 2012), which counted all remembered overnight dreams, reported recall rates of 5.1 ± 3.1 dreams/week for the diary, 6.1 ± 3.5 dreams/week for the checklist, and 3.9 ± 2.1 dreams/week for the questionnaire. Of note, while many studies investigated dream recall frequency using written diaries, the impact of written vs. oral data collection is still unclear. In general, writing is a more demanding task both in terms of preparation and execution, relative to speaking in a voice recorded (Casagrande and Cortini, *Consciousness and Cognition*, 2008). This higher complexity may lead to interference with the act of dream recall itself, leading to dream forgetting, or may negatively affect the participants' motivation level.

The instructions and definitions that are given to participants may also have an important impact on dream recall estimates. As is now well-known, most people think of 'dreams' as the vivid and complex movie-like perceptual experiences typical of REM sleep (Nielsen, *Behav Brain Sci*, 2000). However, in line with the most recent literature (e.g., Siclari et al., *Nat Neurosci*, 2017; Siclari et al., *J Neurosci*, 2019; Stephan et al., *Curr Biol*, 2021), we adopted in our work a broader definition for dreams, and all participants received clear, standardized instructions regarding what they had to consider as dream experiences for their reporting. Specifically, we told them to regard as dreams any subjective conscious experiences, including not only rich and vivid perceptual experiences, but also any perceptual impressions, thoughts, or emotional states, that occurred during sleep, before the morning awakening.

Based on all the above considerations, we are convinced that our apparently high dream recall rate does not reflect a sampling bias (i.e., the recruitment of high dream recallers), but rather a consequence of our specific experimental approach and standardization effort. To further put this into test, we investigated whether attitude towards dreaming (ATD), the strongest predictor of dream recall frequency, showed any clear biases in our sample with respect to previous reports. In particular, since the ATD questionnaire adopted in our work represented a translation of the questionnaire used by Schredl and Bulkeley (Schredl and Bulkeley, *Dreaming*, 2019), we directly compared our scores with the ones reported in their study. Of note, the previous work was performed on a large sample ($N=5,255$) of healthy adult americans.

The ATD questionnaire includes 3 positive and 3 negative statements about dreams. The positive statements are: 1p) *Dreams are a good way of learning about my true feelings*; 2p) *Dreams can anticipate things that happen in the future*; 3p) *Some dreams are caused by powers outside the human mind*. The negative statements are: 1n) *Dreams are random nonsense from the brain*; 2n) *I am too busy in waking life to pay*

attention to my dreams; 3n) I get bored listening to other people talk about their dreams. For each statement, the participants are asked if they *strongly agreed* (4), *somewhat agreed* (3), *neither agreed nor disagreed* (2), *somewhat disagreed* (1), or *strongly disagreed* (0). The table below compares our obtained scores for each statement with the ones of the study by Schredl and Bulkeley (below indicated as ‘reference sample’; Table R1). From this comparison, our sample appeared to have marginally higher scores for positive statement 1p, and lower scores for the three negative statements (1n-3n), but also showed lower scores for positive statements 2p and 3p relative to the reference sample. Overall, the observed differences appear small and could reflect random and/or cultural differences. Indeed, mean values obtained in our study are all within one standard deviation from the mean values reported in the reference sample. As such, this comparison showed no evidence of a systematic bias in our sample.

Statements	Elce et al.	Schredl & Bulkeley
1p - Learning about true feelings	2.74 ± 1.05	2.30 ± 1.09
2p - Dream can anticipate the future	1.02 ± 1.13	2.00 ± 1.24
3p - Dreams can be caused by outside powers	1.27 ± 1.24	1.96 ± 1.24
1n - Getting bored listening to others talk about dreams	0.96 ± 1.15	1.76 ± 1.15
2n - Too busy to pay attention to dreams	1.25 ± 1.23	1.71 ± 1.19
3n - Dreams are random nonsense	0.72 ± 0.99	1.82 ± 1.20

Table R1. Scores attributed to each positive and negative statement of the attitude towards dreaming questionnaire in the present study (left column) and in the reference sample examined in the work by Schredl and Bulkeley (right column).

We modified the main text of the manuscript to incorporate the above observations and considerations, as detailed below.

Results, Page 10, Line 304: “Fig. 1b shows the proportion of morning reports corresponding to CD (mean ± standard deviation = 0.58 ± 0.24), WD (0.14 ± 0.13), and ND (0.28 ± 0.22). On average, CD+WD probability was 0.72 ± 0.22, corresponding to 5.04 ± 1.54 dreams per week, ~~in-line with previous observations concerning dream recall frequency [19]~~”. Here we removed the part of the sentence referring to dream recall frequency in the previous literature and expanded instead the Discussion section on this matter (see below).

Limitations, Page 15, Line 504: “While the dream recall rate reported in this study (CD = 4.1 per week; CD+WD = 5.0 per week) may appear relatively high compared to previous research, commonly pointing to a range of 1-3 recalled dreams per week (e.g., [REF27]; but see also [REF28]), the observed values may be explained by the methodological approach we adopted. For example, while here we instructed participants to consider as dreams any subjective conscious experience occurring during sleep (e.g., [REF29]), other studies allowed participants to apply their own definition of what constitutes a dream, or limitedly focused on perceptual, vivid, and rich (REM-like) experiences. Moreover, while most studies

relied on written morning reports, here we used oral reporting. Importantly, written reports are simpler and faster to process for experimenters but entail a greater effort from participants and might be expected to also more easily determine interference effects [REF30]. Since these and other methodological choices may affect dream recall estimates, we argue that the field would benefit from consensus-level standardization and improvements in methodological reporting to enable more reliable comparisons across studies and further research advancement”.

Methods, Page 15, Line 145: “Volunteers who met all the inclusion criteria were provided with an actigraph and a voice-recorder and were asked to record each morning, upon awakening from sleep, everything that was going through their mind just before they woke up, everything they remembered, every experience or thought they had before awakening. *It is important to note that this procedure differed in several respects from those commonly used in home-based dream experiments. First, all participants received clear, standardized instructions on what to consider as a dream. In particular, we adopted a broad definition, encompassing any subjective conscious experience occurring during sleep [REF29]. Second, while in some studies participants are asked to report all dreams that they have during a given sleep night, here we instructed them to only focus on the very last experience that they had before awakening. This choice was made in order to minimize the impact of confounding effects that may intervene between the dream experience and the report. Third, while most previous studies relied on written reporting, here we used voice recordings, as this approach renders the recall task simpler and less demanding for participants [REF30]. Furthermore, at pseudo-random times during the day, participants were also contacted and asked to record everything that was going through their minds up to 15 minutes before they started the recording. In particular, a simple phone text message containing the word “record” (“registra”) was sent to the volunteers at pseudo-random times during the day (Fig. 1a). Wakefulness reports were not analyzed in the present study”.*

REF27: Schredl, M. (2008). Dream recall frequency in a representative German sample. *Perceptual and Motor Skills, 106*(3), 699-702.

REF28: Schredl, M. (2002). Questionnaires and diaries as research instruments in dream research: Methodological issues. *Dreaming, 12*, 17-26.

REF29: Siclari, F., Baird, B., Perogamvros, L., Bernardi, G., LaRocque, J. J., Riedner, B., ... & Tononi, G. (2017). The neural correlates of dreaming. *Nature neuroscience, 20*(6), 872-878.

REF30: Casagrande, M., & Cortini, P. (2008). Spoken and written dream communication: Differences and methodological aspects. *Consciousness and cognition, 17*(1), 145-158.

8. I have problems with the interpretation of PC2, in my opinion, the main factor is sleep duration, the negative correlation does make sense as longer sleep includes a smaller percentage of N3 sleep (as this sleep stage is lacking in the second part of the night). So the authors should check on studies, looking at the association between sleep duration and dream recall, e.g., Schredl, M., & Fulda, S. (2005). Dream recall and sleep duration: state or trait factor. *Perceptual and Motor Skills, 101*, 613-616. Or Schredl, M., & Reinhard, I. (2008). Dream recall, dream length and sleep duration: state or

trait factor. Perceptual and Motor Skills, 106, 633-636. These findings are somewhat in line with the data presented.

We appreciate the Reviewer's suggestion. However, we note that the studies mentioned by the Reviewer and the more recent work by Tschunichin & Schredl (Tschunichin & Schredl, *Somnologie*, 2024) are all based on self-reported sleep duration and did not assess variations in sleep macro- or micro-structure. As such, these studies were unable to investigate the possible impact of specific changes in sleep structure on dream recall. In this regard, we also note that self-reported sleep duration may not represent a reliable index of actual (PSG-based) sleep duration, as it may also reflect differences in subjective sleep quality (e.g., Lecci et al., *SLEEP*, 2020). More importantly, though, we note that sleep duration and the percentage of N3 sleep may dissociate (e.g., Lecci et al., *SLEEP*, 2020). Given these considerations, while we agree with the Reviewer that sleep duration may contribute to explain the negative correlation between PC2 and N3 sleep, we also believe that sleep duration and N3 amount may not fully overlap. Instead, these variables carry distinct, though inevitably related, information. We hope the Reviewer will agree with our considerations.

To better compare our present results with those of previous investigations, we modified the Discussion section by clarifying that our results are consistent with studies reporting a correlation between dream recall and sleep duration.

Page 13, Line 442: "Specifically, we found that individuals who typically have long sleep episodes coupled with a low proportion of deep N3 sleep, exhibit a heightened probability of dream recall compared to those experiencing shorter, N3-rich sleep. This observation aligns with prior investigations describing a positive correlation between sleep duration and dream recall [REF31] and a negative relationship between the occurrence of slow waves typical of NREM sleep and dreaming [45]"

REF31: Tschunichin, D., Schredl, M. Dream recall, white dreaming, and sleep duration: A diary study in patients with sleep disorders. *Somnologie* (2024). <https://doi.org/10.1007/s11818-024-00479-y>

9. In the section "Seasonal variations", I do not see any Ns, how many participants are in the different group. And I also do not understand why sleep parameters were not controlled for in the analysis of possible seasonal effects on dream recall.

We thank the Reviewer for having pointed our attention towards this issue. The texts indicating sample sizes were not correctly exported when the image was generated. This issue was corrected in the revised Figure 4 (see below, Fig. R3). The analysis included 48 participants in Winter, 50 in Spring, 42 in Summer, and 60 in Autumn.

Seasonal changes in dream recall and sleep

Figure R3. Changes in morning dream recall (a) and sleep patterns reflected by the actigraphy-PC2 (b) across seasons. * $q < 0.05$, ** $q < 0.01$, *** $q < 0.001$ (FDR correction). Dream recall probability values were adjusted for age, sex, vulnerability to interference, attitude towards dreaming, and PC2. PC scores were adjusted for participants' age and sex. *The analysis included 48 participants in Winter, 50 in Spring, 42 in Summer, and 60 in Autumn.*

Recall parameters were adjusted for age, sex, attitude towards dreaming, proneness to mind wandering, and actigraphy-PC2 values, as PC2 was the only actigraphy-based predictor that showed significant effects on dream recall rates. The other PCs were not included because analyses indicated no significant associations for PC1, PC3, or PC4 with dream recall. Additionally, our investigation of seasonal variations in actigraphy-derived sleep patterns found significant changes only for PC2.

To address the Reviewer's interest, we repeated the analysis of seasonal variations in dream recall rates, this time controlling for all actigraphy-derived PCs along with age, sex, attitude towards dreaming, and proneness to mind wandering. The results remained consistent with those reported in the manuscript, showing lower dream recall during Winter compared to Spring ($q = 0.022$, FDR corrected). A non-significant trend was also observed between Winter and Autumn ($p = 0.066$, *uncorrected*).

10. Lastly, the authors have also elicited dream recall frequency (retrospectively). Based on their introduction, it would be very interesting to test whether influencing factors (the whole range) correlate differently for the two measures, and how high the correlation between the two measures are. This might be lower compared to other samples because of the high recaller bias in this study.

We agree that assessing possible differences between dream recall rates derived from the verbal diary and self-reported dream recall frequency may be of interest. In the previous version of the manuscript (page 6) we only reported the mean dream recall frequency derived from the questionnaire (2.66 ± 2.29 dreams per week; computed as in Schredl 2002) and the correlation between the two recall rate measures (Spearman's correlation, $r = 0.46$, $CI = [0.35, 0.55]$; Fig. R4). Of note, this latter value is in line with previous literature investigating the relationship between prospective and retrospective measures of dream recall, which showed a correlation coefficient ranging between $r = 0.33$ and $r = 0.69$ (Nemeth, *Psychological Research*, 2023).

Figure R4. Correlation between dream recall rates measured from the experimental morning reports (*x*-axis; expressed as morning probability of dream recall) and the retrospective questionnaire (*y*-axis; expressed on the questionnaire ordinal scale). Each dot represents a study participant ($N = 204$). A significant correlation was observed between the two measures (Spearman's correlation, $r = 0.46$, $CI = [0.35, 0.55]$).

In response to the Reviewer's request, we conducted new analyses to explore potential predictors of dream recall frequency as assessed by the retrospective questionnaire. Given the nature of the outcome variable, we applied a standard linear mixed-effects model, rather than a generalized linear mixed-effects logistic model, which was used for analyzing dream recall rates based on prospective measures. Due to the model's lower statistical power and the unfavorable ratio of observations to predictors, we performed three separate analyses, each focusing on different sets of variables: cognitive/demographic factors, sleep-related factors, and predictors that showed significant (uncorrected) effects in the original analyses of prospectively measured dream recall rates.

Model #1 - Cognitive and demographic variables. We included as predictors age, sex, education, attitude towards dreaming, vulnerability to interference, verbal memory, visual memory, anxiety, sleep quality, circadian preference, vividness of visual imagery, and proneness to mind wandering. The model revealed an FDR corrected effect only for attitude towards dreaming ($q = 0.013$). However, non-significant trends were also observed for age ($p = 0.026$), sex ($p = 0.092$), verbal memory ($p = 0.020$), and mind wandering ($p = 0.030$).

Model #2 - Sleep-related factors. We included as predictors age, sex, and the four PCs derived from actigraphic data. This analysis revealed no significant effects after FDR correction. However, a non-significant trend was observed for PC2 ($p=0.021$).

Model #3 - Main factors from prospective study. We included as predictors age, sex, education, attitude towards dreaming, vulnerability to interference, verbal memory, proneness to mind wandering, and

actigraphy-PC2. This analysis revealed significant (FDR corrected) effects for attitude towards dreaming ($q = 0.006$), verbal memory ($q = 0.032$), proneness to mind wandering ($q = 0.032$), and PC2 ($q = 0.026$). With the exception of verbal memory, these latter results overlap substantially with those obtained for prospective measures of dream recall rates expressed as the sum of contentful and white dreams.

Name	Estimate	SE	tStat	pValue	Lower CI	Upper CI
Intercept	-0.049626	1.4265	-0.034789	0.97228	-2.8634	2.7642
Sex	-0.18687	0.21118	-0.88488	0.37734	-0.2297	0.6034
Age	-0.013829	0.0091014	-1.5194	0.13032	-1.6209	0.2103
Education	0.0077	0.036798	0.20925	0.83448	-0.9733	1.2043
ATD	0.089554	0.02631	3.4037	0.0008107	0.7155	2.6876
SCWT	0.003296	0.01599	0.20613	0.83691	-1.2233	1.5088
BSRT	0.069586	0.028636	2.4301	0.016026	0.2649	2.5491
MW	0.21232	0.084735	2.5057	0.01306	0.2259	1.8973
PC2	0.23008	0.083509	2.7551	0.006438	0.5213	3.1490

Table R2. LME results for dream recall frequency as derived from the retrospective questionnaire ($N = 199$; $df = 190$; Adjusted $R^2 = 0.14$). This confirmatory analysis was performed using a linear mixed effect model including as predictors the factors that showed significant (uncorrected) effects in the original analyses of prospectively measured dream recall rates. For each predictor, we reported the estimate, standard error (SE), t statistics, p -value, and lower and upper bounds of the effect 95% confidence interval. This analysis revealed statistically significant (FDR corrected) effects for attitude towards dreaming ($q = 0.006$), verbal memory ($q = 0.032$), proneness to mind wandering ($q = 0.032$), and PC2 ($q = 0.026$).

Reviewer #2

The authors describe a large longitudinal study on dream reports. Dream recall was measured at awakening in the morning and sleep was assessed using actigraphy. A subset of participants also used a dry electrode polysomnography device, which was analysed using an automated sleep scoring algorithm. Inter-individual differences were assessed and related to the dream recall and the authors find that attitudes towards dreaming, proneness to mind wandering and long light sleep (a principal component derived from the actigraphy data) were positively related to increased dream recall. Contentful dream recall vs. white dream recall (recall of the feeling of dreaming without knowing the content) was associated with age and vulnerability to interference. Of note, none of the memory measures was related to dream recall. In addition, the authors investigated the relationship between dream recall and seasons. Here they found that dream recall is lower in winter than in spring. Overall the study is interesting and conducted at a high level. I only have a few concerns that should be addressed.

We thank the Reviewer for their positive assessment of our work and their insightful comments.

1. From the methods, it was unclear to me, if the PCA took into account that some of the measurement came from the same individuals, i.e., were dependent. This seems like an important issue of the data structure that would bias the PCA.

We thank the Reviewer for highlighting this potential concern. Indeed, our Principal Component Analysis (PCA) was conducted by combining all actigraphic data (i.e., 24 descriptors) from different nights across the entire sample of participants, effectively mixing both intra- and inter-subject variability. The rationale for this decision was related to the distribution of nights collected across participants. After excluding nights where the actigraph appeared to have been removed, data was lost, or contained missing values, we counted the number of nights with reliable actigraphic information for each subject (Figure R5, panel A). The experiment lasted 15 days, but a few participants extended the duration of the experimental period for different personal or health reasons. Overall, the average number of completed nights per subject was 14.0 ± 1.6 , with a minimum of 8 and a maximum of 17 nights, and with ~72% of the participants having 14 or 15 nights of data. The bar plot in Figure 5B shows for each experimental day, from day 1 to day 19 (reached by two participants), the number of subjects who collected usable data for that specific day. This plot indicates that, within the 15-day period, the number of available observations is relatively stable. This observation suggests that, although we mixed both intra- and inter-subject variability, the amount of collected data was not biased towards specific participants or experimental nights within the planned duration of the experiment.

A**B**
Figure R5. a) Number of experimental nights per participant; b) Amount of nights collected across participants for each day of the experimental period.

However, to completely rule out any potential bias in the generation of the PCA loadings, we repeated the analysis using *Multiple Factor Analysis* (MFA, Escoufier & Pagès, 1990). MFA is a generalization of PCA designed for data organized into multiple blocks, such as different sets of variables collected from the same observations or, as in our case, the same variables measured across different observations (Abdi et al. 2013). For each experimental night, we created a table including 200 rows (i.e., participants) and 24 columns (i.e., actigraphic measures). This process was performed for the first 15 out of 19 nights of the experiment (see Fig. R5B), leading to the exclusion of about 1% of data compared to the full dataset. This was done to minimize missing values in this restructured dataset. We then performed MFA using R (FactoMineR library, Josse & Husson, 2008) with default parameters (5 factors to be estimated, and each quantitative actigraphic variable scaled to unit variance). This approach provided the factor loadings across variables and for each one of the 15 experimental nights. Since both MFA and PCA can result in flipped loadings, we first applied a Procrustes analysis (with rotation and reflection and without scaling) to align PCA- and MFA-based loadings before measuring their correlations. Finally, we measured the similarity (Pearson's correlation coefficient) between each PC loadings reported in the original version of the manuscript and the corresponding MFA loadings for each one of the 15 nights (see Fig. R6).

Stability of PCs between the canonical PCA and MFA

Figure R6. Pearson correlation between the first 4 PC and MFA loadings. Each line represents the linear fit between the PCA and MFA loadings for each of the 15 experimental nights. The dots correspond to estimates of the MFA and PCA loadings across the 24 actigraphy variables for the first 4 PCs/Factors.

We found that the two techniques produced highly similar loadings across the first four PCs, with average correlations across nights of 0.995 ± 0.004 for PC1, 0.976 ± 0.018 for PC2, 0.986 ± 0.010 for PC3, and 0.924 ± 0.028 for PC4. Overall, this result demonstrates that intra-subject variability did not introduce any bias in our estimation of PC loadings and scores.

Given this evidence, and the key advantage of canonical PCA in generalizing to new data, by projecting it into the PC-defined space through matrix multiplication between the estimated loadings and new actigraphic data, we have opted to retain the use of PCA in the revised version of the manuscript.

2. I am aware of one study showing acceptable performance of the DREEM headband, however, there are considerable reservations in the field of sleep research regarding its validity as a polysomnography device. This should be added as a limitation. In the supplement, the authors should provide example data for good, medium and poor quality data in their sample. In addition, I would like to see the performance of the DREEM algorithm versus another automated scoring (i.e.,

USLEEP, Perselev et al, 2021). Ideally, the authors also show scoring of a subset of the data by experienced technicians.

We agree with the Reviewer that currently available sleep headbands for home recordings, including the DREEM headband, do not have the same reliability and accuracy of laboratory PSG. However, they currently represent the best viable compromise for longitudinal studies aimed at collecting data in more naturalistic conditions.

Please note that one operator manually checked the sleep scoring output and eliminated nights containing obvious issues related to possible device removal or malfunctioning. This allowed us to ensure only the inclusion of recordings and scoring outputs having relatively good quality and reliability. Moreover, as detailed in our response to comment #3, we took the opportunity of this revision to add new automated data quality checks. We added a description of these procedures in the **Method section, Page 7, Line 227**, as follows: *“Data collected using the DREEM device were analyzed using the associated automated sleep scoring software [53]. Of note, one experimenter manually inspected the sleep scoring output and discarded nights containing obvious issues related to possible device removal or malfunction. Moreover, we excluded nights for which less than 5 hours of sleep were recorded, and nights in which more than 25% of all the epochs were marked as unscorable. The obtained hypnograms were used to compute the percentages of wakefulness, N1, N2, N3, and REM sleep for each night (N = 480)”*.

Concerning the Reviewer’s request to validate the DREEM scoring against human operators or other automated scoring algorithms, we regret to note that the results of such a comparison using our data would not be informative. The DREEM headband acquires and analyzes signals from 5 EEG channels sampled at 250 Hz (FPz, F7, F8, O1, O2; from which 7 bipolar derivations could be obtained: FPz-O1, FPz-O2, FPz-F7, F8-F7, F7-O1, F8-O2, FPz-F8), 3 accelerometer channels (sampled at 50hz) and 2 pulse oximeter channels signals (sampled at 50hz). These signals are not among those commonly used in standard sleep scoring either performed by technicians (Iber, AASM, 2007) or automated algorithms (Perselev et al., NPJ Digital Medicine, 2021). Indeed, for instance, the DREEM lacks proper EOG and EMG channels commonly used for scoring REM sleep, but may nevertheless achieve a good scoring accuracy by relying on other, non-conventional signals. Given the above considerations, we are convinced that a proper validation of the DREEM headband could not be performed without a direct comparison against a standard, co-recorded PSG. Importantly, we found three previous studies that performed such a validation (Arnal et al., SLEEP, 2020; González et al., Sleep Health, 2024; Ong et al., Sleep Health, 2024). These studies obtained a very good agreement (> 70%) between the DREEM-derived sleep scoring and the sleep scoring performed by human raters using PSG data.

For the same reasons indicated above, we believe that showing the raw EEG data of a few non-conventional derivations would be of scarce usability and interpretability for the reader. Nevertheless, in keeping with the Reviewer’s suggestion to show recordings with different quality, we included in supplementary materials representative nights including scoring outputs, DREEM accelerometer and actigraphic data (see below Figures R5, R6, and R7; please also see our response to comment #3). It is our hope that these will help the reader to evaluate the relative reliability of DREEM-derived metrics employed in our analyses.

Figure R5. Representative data from good quality data from one participant (DL005). The first row shows the sleep scoring obtained from the DREEM data. The second row shows the accelerometer data derived from the DREEM device. To facilitate comparisons, accelerometer time series values were normalized to a range of 0 (min) to 1 (max). The last row shows the aligned accelerometer data derived from the actigraphy.

Figure R6. Representative data from medium quality data from one participant (DL014). The first row shows the sleep scoring obtained from the DREEM data. The second row shows the accelerometer data derived from the DREEM device. To facilitate comparisons, accelerometer time series values were normalized to a range of 0 (min) to 1 (max). The last row shows the aligned accelerometer data derived from the actigraphy. Here some epochs in the 6th and 7th hour of recording were marked as unscorable.

Figure R7. Representative data from low quality data from one participant (DL006). The first row shows the sleep scoring obtained from the DREEM data. The second row shows the accelerometer data derived from the DREEM device. The last row shows the aligned accelerometer data derived from the actigraphy. To facilitate comparisons, accelerometer time series values were normalized to a range of 0 (min) to 1 (max). Here several epochs after the second hour of sleep were marked as unscorable (these epochs represented less than 25% of all sleep epochs), probably due to a high amount of body movements during the night. Cases like these were rare in our analyzed sample.

A discussion of the limitations related to the use of the DREEM headband was added to the manuscript in the newly added **Limitations section, Page 15, Line 517**, as follows: *“In this work, objective information about sleep patterns was obtained using actigraphic devices and a portable electrophysiological recording system (the DREEM headband). While these instruments may offer valuable information about sleep duration, quality, and structure, they are known to have a lower reliability with respect to standard laboratory polysomnography. At the same time they currently represent the best viable compromise for longitudinal studies aimed at collecting data in naturalistic conditions and in large samples. In the context of the present study, both manual and automatic approaches were used to identify and exclude recordings related to clear device removal or malfunctioning. Moreover, cross-comparisons across actigraphy and DREEM recordings showed a high level of consistency, providing further support to the validity of estimates derived from these devices (see Supplementary Materials Figures S2-4).”*

3. As far as I understand the relationships between actigraphy and DREEM data were made on the aggregate level, i.e., values for entire nights of sleep. To check the validity of these relationships the authors should compare the time series of the actigraphy data to the DREEM data.

The Reviewer is correct. We assessed whether specific sleep macrostructural indices (the percent amount of each sleep stage) of a given night showed any associations with the loadings of each actigraphy-derived

PC computed for the same night. Both the sleep structure and the actigraphy indices were determined for the entire nights of sleep.

As suggested by the Reviewer, we investigated the correspondence between DREEM and actigraphy time series. As mentioned above, the DREEM system includes a tri-axial accelerometer that is used to derive information about respiration. Therefore, we directly compared the accelerometer time series obtained from the DREEM (root-sum-of-squares across the three axes) with that obtained from the actigraphy. For this comparison, accelerometer values were computed for each 30 second epoch of the sleep time (as determined from the DREEM data). The DREEM accelerometer signal was acquired with a 50 Hz sampling rate. Thus, we computed the mean activity values across sampled time points belonging to the same scoring epoch. Instead, the actigraphy data was estimated over 60 second intervals. In this case, the signal was upsampled to match the epoch length used for sleep scoring. To account for possible misalignments between the DREEM and actigraphy time series, we applied the following steps. First, we inspected the time-stamps of the DREEM recordings and corrected obvious time errors, such as those caused by daylight saving time changes. Then, we used cross-correlation to identify the correlation peak between the DREEM and actigraphy time series in a ± 10 minute time range (40 time points), and then shifted the actigraphy time-series accordingly. The actigraphy time series was cut to match the length of the DREEM time series (see Fig. R5-7). Finally, we computed the correlation across time series for each night and study participant.

We found that the DREEM and the actigraphy time series had a moderate to high correlation (Pearson's correlation coefficient, $r = 0.50 \pm 0.10$). While not exceptionally strong, probably due to the different positions of the accelerometers on the sleepers' bodies, the observed correlation indicates a fairly good level of agreement and correspondence between the two signals.

To further assess the validity of the alignment between actigraphy and DREEM data, we evaluated and compared the amount of estimated activity (body movement) in each sleep stage (Fig. 8). In line with previous work (e.g., Giganti et al., 2008; Ibrahim et al., 2003; Wilde-Frenz and Shulz, 1983), we found that body movements decreased from wakefulness to deep NREM sleep (N3), while overall activity was more similar between REM and N2 sleep. The consistency of these results with previous observations and the very low variance observed in the different sleep stages further corroborate the validity and accuracy of the automated DREEM-based sleep scoring, as well as the optimal alignment between actigraphic and electrophysiological data.

Body movements per sleep stage

Figure R8. Mean accelerometer activity (as derived from actigraphic data) for each sleep stage defined according to the automated DREEM sleep scoring. The amount of activity differed significantly across all pairs of sleep stages. * $q < 0.05$, ** $q < 0.01$, *** $q < 0.001$.

The above analyses and results were added as supplementary materials in the revised manuscript (Supplementary Fig. S5). Please note that we took the opportunity of this revision to improve our analyses of the DREEM-based data by adding new quality checks and data exclusion criteria. Specifically, we excluded nights for which less than 5 hours of sleep were recorded, and nights in which more than 25% of all the epochs were marked as unscorable. These criteria led to the exclusion of three nights (in addition to those already excluded based on manual assessments), leading to a total of 480 retained nights. This information was added in the Methods section of the revised manuscript. Of note, the described change had a minimal impact on results. Below we report for the Reviewer's convenience the updated text of the **Results section, Page 10, Line 324**:

“To facilitate the interpretation of the PCs, we employed mixed-effect models including sleep-structure measures obtained in the subsample of participants who wore the portable EEG system during the experimental nights (Tables S2-5). We found that PC1 was positively associated with the proportion of wakefulness ($q = 0.001$, False Discovery Rate -FDR- correction) and N1 ($q = 0.001$; model adjusted $R^2 = 0.47$). Moreover, PC2 was negatively associated with the proportion of N3 sleep ($q < 0.001$; model adjusted $R^2 = 0.41$), whereas PC4 was negatively associated with the proportions of N2, N3, and REM sleep ($q < 0.005$; model adjusted $R^2 = 0.38$). No significant predictors were identified for PC3.”

4. I was somewhat surprised about the frequency of contentful dreams. Although the authors report this to be in line with previous research, I wonder how representative the sample is. How did the authors recruit participants and is it possible that they had an increased interest in dreams at the outset. If this is the case, how does this limit the research?

Participants were recruited through word of mouth and through the dissemination of virtual and paper flyers (this information was added to the manuscript; see below). However, given the obvious risks of especially attracting volunteers with a specific interest in dreams, we mostly relied on word of mouth to reach diverse participants showing different degrees of interest towards dreaming.

We believe that the main reason for our relatively high rate of reported contentful dreams lies in the specific and detailed instructions and definitions that we provided to our participants. Indeed, when individuals are asked to report their dreams, they often interpret this request as one to report (REM-like) vivid and complex perceptual experiences (Nielsen, *Behav Brain Sci*, 2000). However, in line with the most recent literature (e.g., Siclari et al., *Nat Neurosci*, 2017), all participants received clear, standardized instructions regarding what they had to consider as dream experiences for their reporting. Specifically, we told them to regard as dreams any subjective experiences, including not only rich and vivid perceptual experiences, but also any perceptual impressions, thoughts, or emotional states, that occurred during sleep, before the morning awakening. For more in-depth discussions and analyses regarding possible biases in our sample - including an assessment of attitude towards dreaming in our and previous investigations - we invite the Reviewer to read our response to comment #7 of Reviewer #1.

Information regarding how recruitment was performed has been added to the *Methods section, Page 11, Line 371*, as follows: “*Data collection was carried out between March 2020 and March 2024, covering a period of four years. Participants were recruited through word of mouth and the dissemination of virtual and paper flyers. Given the risks of recruiting a majority of volunteers with a specific interest in dreams, we mainly relied on word of mouth to reach diverse participants regardless of their interest towards dreaming*”.

5. In general the figures are informative, but quite hard to read. The asterisks are too small and overall the size of graphical elements could be optimised. This is especially the case for Figure 1c.

All text and markers in figures have been increased in size to improve readability. Concerning Figure 1c, please also see our response to comment #5 of Reviewer #1.

References

- Abdi, H., Williams, L. J., & Valentin, D. (2013). Multiple factor analysis: principal component analysis for multitable and multiblock data sets. *Wiley Interdisciplinary reviews: computational statistics*, 5(2), 149-179.
- Arnal, P. J., Thorey, V., Debellemaniere, E., Ballard, M. E., Bou Hernandez, A., Guillot, A., ... & Sauvet, F. (2020). The Dreem Headband compared to polysomnography for electroencephalographic signal acquisition and sleep staging. *Sleep*, 43(11), zsaa097.
- Bulkeley, K., & Schredl, M. (2019). Attitudes towards dreaming: Effects of socio-demographic and religious variables in an American sample. *International Journal of Dream Research*, 75-81.
- Casagrande, M., & Cortini, P. (2008). Spoken and written dream communication: Differences and methodological aspects. *Consciousness and cognition*, 17(1), 145-158.
- Escofier, B., & Pages, J. (1994). Multiple factor analysis (AFMULT package). *Computational statistics & data analysis*, 18(1), 121-140.
- Giganti, F., Ficca, G., Gori, S., & Salzarulo, P. (2008). Body movements during night sleep and their relationship with sleep stages are further modified in very old subjects. *Brain research bulletin*, 75(1), 66-69.
- González, D. A., Wang, D., Pollet, E., Velarde, A., Horn, S., Coss, P., ... & Gonzales, M. M. (2024). Performance of the Dreem 2 EEG headband, relative to polysomnography, for assessing sleep in Parkinson's disease. *Sleep Health*, 10(1), 24-30.
- Iber, C. (2007). The AASM manual for the scoring of sleep and associated events: Rules. *Terminology and Technical Specification*.
- Ibrahim, A., Ferri, R., Cesari, M., Frauscher, B., Heidbreder, A., Bergmann, M., ... & Stefani, A. (2023). Large muscle group movements during sleep in healthy people: normative values and correlation to sleep features. *Sleep*, 46(8), zsad129.
- Lê, S., Josse, J., & Husson, F. (2008). FactoMineR: an R package for multivariate analysis. *Journal of statistical software*, 25, 1-18.
- Lecci, S., Cataldi, J., Betta, M., Bernardi, G., Heinzer, R., & Siclari, F. (2020). Electroencephalographic changes associated with subjective under-and overestimation of sleep duration. *Sleep*, 43(11), zsaa094.
- Nemeth, G. (2023). The route to recall a dream: Theoretical considerations and methodological implications. *Psychological Research*, 87(4), 964-987.
- Nielsen, T. A. (2000). A review of mentation in REM and NREM sleep: "covert" REM sleep as a possible reconciliation of two opposing models. *Behavioral and Brain Sciences*, 23(6), 851-866.

- Nielsen, T. A. (2000). Covert REM sleep effects on REM mentation: Further methodological considerations and supporting evidence. *Behavioral and Brain Sciences*, 23(6), 1040-1057.
- Ong, J. L., Golkashani, H. A., Ghorbani, S., Wong, K. F., Chee, N. I., Willoughby, A. R., & Chee, M. W. (2024). Selecting a sleep tracker from EEG-based, iteratively improved, low-cost multisensor, and actigraphy-only devices. *Sleep Health*, 10(1), 9-23.
- Perslev, M., Darkner, S., Kempfner, L., Nikolic, M., Jennum, P. J., & Igel, C. (2021). U-Sleep: resilient high-frequency sleep staging. *NPJ digital medicine*, 4(1), 72.
- Schredl, M. (2002). Questionnaires and diaries as research instruments in dream research: Methodological issues. *Dreaming*, 12, 17-26.
- Siclari, F., Baird, B., Perogamvros, L., Bernardi, G., LaRocque, J. J., Riedner, B., ... & Tononi, G. (2017). The neural correlates of dreaming. *Nature neuroscience*, 20(6), 872-878.
- Siclari, F., LaRocque, J. J., Postle, B. R., & Tononi, G. (2013). Assessing sleep consciousness within subjects using a serial awakening paradigm. *Frontiers in psychology*, 4, 542.
- Stephan, A. M., Lecci, S., Cataldi, J., & Siclari, F. (2021). Conscious experiences and high-density EEG patterns predicting subjective sleep depth. *Current Biology*, 31(24), 5487-5500.
- Tschunichin, D., & Schredl, M. (2024). Dream recall, white dreaming, and sleep duration: A diary study in patients with sleep disorders. *Somnologie*, 1-5.
- Wilde-Frenz, J., & Schulz, H. (1983). Rate and distribution of body movements during sleep in humans. *Perceptual and motor skills*, 56(1), 275-283.
- Wood, E., Westphal, J. K., & Lerner, I. (2023). Re-evaluating two popular EEG-based mobile sleep-monitoring devices for home use. *Journal of Sleep Research*, 32(5), e13824.
- Zadra, A., & Robert, G. (2012). Dream recall frequency: Impact of prospective measures and motivational factors. *Consciousness and Cognition*, 21(4), 1695-1702.

Please find below our point-by-point response to each comment and a description of all changes made to the manuscript. For the Reviewers' convenience, we reported in our responses the portions of the manuscript that we revised, highlighting changed text in red color. Please note that we took the opportunity of this revision to correct some typos and a few display errors in Figure 2. We also corrected a few minor copy-pasting errors in supplementary tables.

Reviewer #2

The authors have substantially improved the manuscript. However, there remain two points. The first is just to give them credit for actually finding a good way to deal with structured data and should strengthen the ms. without a lot of work. The second is a crucial point that should have been addressed in the first revision.

We thank the Reviewer for acknowledging the improvements made to the manuscript and for their constructive feedback.

Reviewer #2 my original point 1: This is convincing. Please add a sentence to the manuscript that this was done and report the correlations between MFA and PCA.

The following text was added to the manuscript (Methods section, Line 234): *“The PCA was applied by combining all actigraphic data from different nights across the entire sample of participants, effectively mixing intra- and inter-subject variability. In order to rule out potential biases in the generation of the PCA loadings, we repeated the analysis using Multiple Factor Analysis (MFA, [REF1]). MFA is a generalization of PCA designed for data organized into multiple blocks, such as different sets of variables collected from the same observations or, as in our case, the same variables measured across different observations [REF2]. We found that the two techniques produced highly similar loadings across the first four PCs, with average correlations across nights of 0.995 ± 0.004 for PC1, 0.976 ± 0.018 for PC2, 0.986 ± 0.010 for PC3, and 0.924 ± 0.028 for PC4 (Supplementary Fig. S1). Overall, this evidence demonstrates that intra-individual variability did not introduce any bias in our estimation of PC loadings and scores.”*

Moreover, Figure R6 of our previous response was added to Supplementary Materials (Figure S1), with the following caption: *“Pearson correlations between the first four principal components (PCs) and MFA loadings. For each of the first 15 experimental nights, we generated a table with 200 rows (participants) and 24 columns (actigraphy measures), excluding ~1% of data from participants with extended experiment durations to minimize missing values. MFA was performed in R (FactoMineR library; [REF3]) using default parameters (5 factors). Quantitative actigraphic variables were scaled to unit variance. Procrustes analysis was applied to align PCA and MFA loadings (allowing rotation and reflection) before measuring correlations. Each line in the plots represents the linear fit between PCA and MFA loadings for one night,*

with dots showing individual loading estimates for the 24 actigraphy variables across the first four PCs/factors.”

Ref1: Escofier, B. & Pagès, J. Multiple factor analysis. *Comput. Stat. Data Anal.* **18**, 120–140 (1990).

Ref2: Abdi, H., Williams, L. J. & Valentin, D. Multiple factor analysis: principal component analysis for multitable and multiblock data sets. *WIREs Computational Statistics* **5**, 149–179 (2013).

Ref3: Lê, S., Josse, J., & Husson, F. FactoMineR: an R package for multivariate analysis. *Journal of statistical software* **25**, 1–18 (2008).

Reviewer #2 my original point 2: I have close to 20 years of experience in sleep research. In this time I have learned enough about sleep-EEG that seeing some raw data will be informative irrespective of the derivation. This is also the case for many of our colleagues. Therefore, I ask the authors to provide this in the supplement. USLEEP is able to handle a broad variety of derivations and it would be informative to see how it performs, even if the ground truth remains elusive. I do not have confidence in the DREEM data unless this is added to the manuscript.

Following the Reviewer’s request, we compared sleep scoring outputs from DREEM and USLEEP. Representative results from the three participants and nights presented in our earlier rebuttal letter (representing good, fair, and poor data quality) are provided in Figures R1, R2, and R3 (Panel A), along with example EEG traces for N2, N3, and REM epochs (Panel B; the representative hypnograms and EEG traces have been also included in the article Supplementary Materials as Figures S4, S6, and S8). The presented results are consistent with those obtained from other participants and nights, as well as across different USLEEP input configurations (e.g., including only F7-F8 as the EOG channel). Results presented here were obtained using USLEEP v2.0, with the following EEG and EOG channels as input: EEG = Fpz-O1, Fpz-O2, F7-O1, F8-O2; EOG = Fpz-F7, Fpz-F8, F7-F8. Our analysis of a data subsample revealed that the agreement between DREEM and USLEEP scoring typically ranged from low to moderate, varying across different cases. For instance, the three participants and nights included in our previous response achieved concordance rates of 47.5%, 37.0%, and 27.5%. Of note, USLEEP scored epochs deemed ‘unscorable’ by DREEM, often with confidence levels similar to other epochs. This issue was evident in participants DL014 and DL006 (i.e., those with the lowest agreement), as their nights included several unscorable epochs. Importantly, USLEEP confidence levels were often low (around or below 0.5), and the generated hypnograms appeared less plausible in terms of sleep stage distribution and cycling compared to DREEM-based ones. As a scoring quality check, we assessed actigraphy-based activity distributions across sleep stages for both algorithms (Figure R4). DREEM-derived distributions aligned more closely with published findings and our group-level results (Supplementary Fig. 9; also see our response to point #3 in the previous rebuttal letter).

Without a gold standard (human-scored polysomnography), it is not possible to definitively identify which algorithm is more accurate. However, the above observations suggest that USLEEP may struggle to accurately score DREEM-headband data. This is not surprising given that: i) the currently available and validated version of USLEEP requires at least one standard EOG channel, which is absent in DREEM recordings; ii) the DREEM-based scoring algorithm uses non-EEG signals (i.e., accelerometry, photoplethysmography) that may ensure scoring accuracy when EEG-signal quality is reduced - this is not

the case for USLEEP. Crucially, USLEEP has not been validated on DREEM data, whereas DREEM has demonstrated acceptable accuracy and reliability against polysomnography in three published studies (Arnal et al., SLEEP, 2020; González et al., Sleep Health, 2024; Ong et al., Sleep Health, 2024).

We believe that comparing DREEM and USLEEP outputs without a gold standard offers limited value and risks leading to misleading interpretations. As noted in our previous response and emphasized in the Limitations section of the revised manuscript, we acknowledge that laboratory-grade PSG would be necessary to confirm the findings obtained using the DREEM headband and its sleep-scoring algorithm. However, it is important to highlight that this aspect of the analysis primarily relates to the interpretation of actigraphy-derived principal components (PCs) and does not impact the main findings or conclusions of our study.

Figure R1

Figure R1. Comparison between DREAM (first row) and USLEEP (second row) scoring for a representative night of participant DL005. The third row shows epochs that were scored differently by the two algorithms (gray bars) and the confidence level of USLEEP (green line; smoothed with a 5-point moving average). The three EEG (μV) epochs shown in panel B (circled on the sleep scoring in panel A) were scored by USLEEP as N2, N3, and N2, respectively.

Figure R2

Figure R2. Comparison between DREEM (first row) and USLEEP (second row) scoring for a representative night of participant DL014. The third row shows epochs that were scored differently by the two algorithms (gray bars) and the confidence level of USLEEP (green line; smoothed with a 5-point moving average). The three EEG (μV) epochs shown in panel B (circled on the sleep scoring in panel A) were scored by USLEEP as W, REM, and N3, respectively.

Figure R3

Figure R4

Figure R4. Mean accelerometer activity (as derived from actigraphic data) for each sleep stage as defined according to the automated DREEM sleep scoring (red) and the USLEEP algorithm (blue), for the four representative nights shown in Figures R1, R2, R3.

References

Arnal, P. J., Thorey, V., Debellemanni, E., Ballard, M. E., Bou Hernandez, A., Guillot, A., ... & Sauvet, F. The Dreem Headband compared to polysomnography for electroencephalographic signal acquisition and sleep staging. *Sleep*, **43(11)**, (2020).

González, D. A., Wang, D., Pollet, E., Velarde, A., Horn, S., Coss, P., ... & Gonzales, M. M. Performance of the Dreem 2 EEG headband, relative to polysomnography, for assessing sleep in Parkinson's disease. *Sleep Health*, **10(1)**, 24-30 (2024).

Ong, J. L., Golkashani, H. A., Ghorbani, S., Wong, K. F., Chee, N. I., Willoughby, A. R., & Chee, M. W. Selecting a sleep tracker from EEG-based, iteratively improved, low-cost multisensor, and actigraphy-only devices. *Sleep Health*, **10(1)**, 9-23 (2024).